# Dynamin independent endocytosis is an alternative cell entry mechanism for multiple animal viruses

**Ravi Ojha**[1�], **Anmin Jiang**[2�], **Elina Mäntylä**[3], **Tania Quirin**[1], **Naphak Modhira**[4], **Robert Witte**[5], **Arnaud Gaudin**[6], **Lisa De Zanetti**[1,7], **Rachel Sarah Gormal**[2], **Maija Vihinen-Ranta**[8], **Jason Mercer**[9], **Maarit Suomalainen**[5], **Urs F. Greber**[5], **Yohei Yamauchi**[10], **Pierre-Yves Lozach**[11], **Ari Helenius**[12], **Olli Vapalahti**[1,13,14], **Paul Young**[4], **Daniel Watterson**[4], **Frédéric A. Meunier**[2,4,6]\*, **Merja Joensuu**[2,15]\*, **Giuseppe Balistreri**[1,6]\*

1 Department of Virology, Faculty of Medicine, University of Helsinki, Helsinki, Finland, 2 Clem Jones Centre for Ageing Dementia Research, Queensland Brain Institute, The University of Queensland, Brisbane, Queensland, Australia, 3 Faculty of Medicine and Health Technology, Tampere University, Tampere, Finland, 4 School of Biomedical Sciences, The University of Queensland, St Lucia, Queensland, Australia, 5 Department of Molecular Life Sciences, University of Zurich, Zurich, Switzerland, 6 Queensland Brain Institute, The University of Queensland, Brisbane, Queensland, Australia, 7 Laboratory of Functional Plant Biology, Department of Biology, Ghent University, Ghent, Belgium, 8 Department of Biological and Environmental Science, and Nanoscience Center, University of Jyvaskyla, Jyvaskyla, Finland, 9 Institute of Microbiology and Infection, School of Biosciences, University of Birmingham, Birmingham, United Kingdom, 10 Institute of Pharmaceutical Sciences, ETH Zurich, Zurich, Switzerland, 11 IVPC UMR754, INRAE, Universite Claude Bernard Lyon 1, EPHE, PSL Research University, Lyon, France, 12 Department of Biochemistry, ETH Zurich, Zurich, Switzerland, 13 Helsinki University Hospital, Helsinki, Finland, 14 Department of Veterinary Biosciences, University of Helsinki, Helsinki, Finland, 15 Australian Institute for Bioengineering and Nanotechnology, The University of Queensland, Brisbane, Queensland, Australia

½ These authors contributed equally to this work.

\* f.meunier@uq.edu.au (FAM); m.joensuu@uq.edu.au (MJ); giuseppe.balistreri@helsinki.fi (GB)

**Data Availability Statement:** All relevant data are within the manuscript and its Supporting Information files.

## Abstract

Mammalian receptor-mediated endocytosis (RME) often involves at least one of three isoforms of the large GTPase dynamin (Dyn). Dyn pinches-off vesicles at the plasma membrane and mediates uptake of many viruses, although some viruses directly penetrate the plasma membrane. RME is classically interrogated by genetic and pharmacological interference, but this has been hampered by undesired effects. Here we studied virus entry in conditional genetic knock-out (KO) mouse embryonic fibroblasts lacking expression of all three dynamin isoforms (Dyn-KO-MEFs). The small canine parvovirus known to use a single receptor, transferrin receptor, strictly depended on dynamin. Larger viruses or viruses known to use multiple receptors, including alphaviruses, influenza, vesicular stomatitis, bunya, adeno, vaccinia, severe acute respiratory syndrome coronavirus 2 (SARS-CoV-2) and rhinoviruses infected Dyn-KO-MEFs, albeit at higher dosage than wild-type MEFs. In absence of the transmembrane protease serine subtype 2 (TMPRSS2), which normally activates the SARS-CoV-2 spike protein for plasma membrane fusion, SARS-CoV-2 infected angiotensin-converting enzyme 2 (ACE2)-expressing MEFs predominantly through dynamin- and actin-dependent endocytosis. In presence of TMPRSS2 the ancestral Wuhan-strain bypassed both dynamin-dependent and -independent endocytosis, and was less sensitive to endosome maturation inhibitors than the Omicron B1 and XBB variants, supporting

**Funding:** For the funding that supported this research we thank the University of Helsinki Graduate Program in Microbiology and Biotechnology for supporting R.O (2019-2023). The strategic Research Council of Finland grant 335527 (G.B), The Sigrid Juselius Foundation (G.B., 2024-2028 https://www.sigridjuselius.fi/en/), the Faculty of Medicine at the University of Helsinki (G.B.), The Helsinki Institute for Life Sciences at University of Helsinki (2023-2025, G.B.) the Jane and Aatos Erkko Foundation (O.P.V., https://jaes.fi/en/frontpage), Helsinki University Hospital funds TYH2021343 (O.P.V.), European Union's Horizon Europe Research and Innovation Program grant 101057553 (G.B., O.P.V.), The University of Queensland Amplify Fellowship (M.J.), the Australian National Health and Medical Research council (grant 2010917 G.B. and F.A.M.). This work was also supported by Australian Research Council (ARC) Discovery Early Career Researcher Award (DE190100565). The work was supported by an ARC Linkage Infrastructure Equipment and Facilities grant (LE130100078) and a National Health and Medical Research Council (NHMRC) Senior Research Fellowship (GNT1155794) to F.A.M.; P.Y.L is supported by the Agence Nationale de la Recherche (ANR) funding (grant numbers ANR-21-CE11-0012 and ANR-22-CE15-0034). Work in the Greber lab was supported by grants from Schweizerischer Nationalfonds (Swiss National Science Foundation 31003A_179256 and 310030_212802). Y.Y. was supported by ERC Synergy grant CHUbVi (ID: 856581). We are grateful to the Jane and Aatos Erkko Foundation for support to M.V.R. and the Research Council of Finland grant 330896 (to M.V.R.) and 332615 (to E.M.). J.M. was supported by core funding to MRC Laboratory for Molecular Cell Biology at University College London (MC_UU_00012/7). The funders had no role in study design, data collection and analysis, decision to publish, or preparation of the manuscript.

**Competing interests:** The authors have declared that no competing interests exist.

the notion that the Omicron variants do not efficiently use TMPRSS2. Collectively, our study suggests that dynamin function at endocytic pits can be essential for infection with single-receptor viruses, while it is not essential but increases uptake and infection efficiency of multi-receptor viruses that otherwise rely on a functional actin network for infection.

## Author summary

To initiate their infection cycle, most viruses first need to enter their target cells, a process called endocytosis. In mammalian cells, endocytosis often involves a class of proteins called dynamins. While numerous viruses, including SARS-CoV-2, efficiently infect cells by dynamin-mediated endocytosis we found that in the absence of these proteins an alternative cell entry mechanism exists, allowing multiple pathogenic human viruses to enter and infect cells. Unlike the dynamin-mediated infection, the efficient internalization of viral particles via dynamin-independent endocytosis seems to always require functional actin fibers, which are structural components of the cell contractile cytoskeleton. Thus, multiple viruses can infect their target cells using at last two entry mechanisms, inhibiting both may provide effective antiviral therapies.

## Introduction

For their replication in the cytoplasm or the nucleus, most animal viruses use receptor-mediated endocytosis (RME) pathways [1] through which the cells engulf particles and nutrients into vesicular compartments. Multiple endocytic mechanisms have been described and viruses have been shown to utilize one or more of these pathways [2,3]. In addition to endocytosis, some viruses infect cells by penetrating directly through the plasma membrane (PM) [4] demonstrating the remarkably fexibility of viruses and their interactions with a diverse range of host cells. How viruses utilise endocytic processes is still incompletely known, but it is key to understanding viral tropism for different cell types, and opening of potential cellular targets for infection interference. Viral endocytosis starts by the virus particle engaging a receptor at the plasma membrane [3], often times followed by viral motions driven by the host receptors and the underlying acto-myosin cytoskeleton, membrane deformation and pinching off of a vesicle containing the virus particle to the cytoplasm [5]. The fission process requires energy typically provided by one of the three dynamin isoforms (dyn1-3), all of which have GTPase activity [6]. Although not all the endocytic pathways require dynamin and some use the actin cytoskeleton, membrane deforming proteins, and motor proteins to generate endocytic vesicles [7–9], dyn-dependent pathways have been the predominant ones described for mammalian viruses [2]. While some viruses, such as Vaccinia virus (VACV) [10,11], Lymphotropic choromeningitis virus (LCMV) [12,13], and human papilloma virus (HPV) [14] have been shown to enter cells via dynamin-independent endocytic mechanisms, others such as Semliki Forest virus (SFV) [15,16] and Influenza virus [17,18] have been shown to enter cells via dynamin-dependent as well as dynamin independent endocytosis. The studies addressing the role of dynamins in virus infection have so far being carried using small molecule inhibitors of dynamins, overexpressing dominant-negative mutants of dynamin, or RNA interference (RNAi) approaches. These loss-of-function approaches, although easy to implement, can suffer from uncharacterized off-targets effects [19,20] or incomplete depletion of the dynamin isoforms.

Irrespective of the upstream steps, viruses in endocytic vesicles are sorted to intracellular membranous organelles, from which the free or capsid enclosed viral genome is released to the cytosol, transported to the site of replication or protein synthesis, or becomes subject to antiviral sensing and degradation (for reviews, see [21–25]).

Enveloped viruses, i.e., viruses surrounded by lipid membranes, deliver the viral genome into the cytoplasm by the fusion of the viral and cellular membranes, a process driven by viral surface proteins (often referred to as 'spikes' on the virion). For most enveloped viruses the cue that triggers fusion is the drop in pH that occurs once the viral particle reaches the lumen of endosomal vesicles (e.g., Influenza virus, Vesicular stomatitis virus, Semliki Forest virus, among others) [15]. For other enveloped viruses (including coronaviruses), the fusion is triggered by proteolytic cleavage of the spike proteins which results in conformational changes, the insertion of the viral spike into the host membrane and the fusion of viral and cellular membranes [26,27]. In the case of coronaviruses, these proteolytic cleavages are catalyzed by cellular proteases present either in the endo-lysosomal compartment (e.g., the cysteine protease cathepsin-L) or at the PM (e.g., the transmembrane serine protease 2, TMPRSS2) [28–31]. Depending on the availability of these proteases, virus fusion and the delivery of the viral genome into the cytoplasm can either occur at the PM or early/recycling endosomes (i.e., in cells that express PM serine proteases) or from within endo-lysosomes (i.e., in cells that express active cathepsins in endo/lysosomes but not serine proteases at the cell surface and early endosomes) [32]. The virus requires endocytosis to reach endosomes and lysosomes. Unlike the cell entry in the cells of the respiratory mucosa, the endosomal route of cell entry has been proposed for SARS-CoV-2 infection of human neurons, which do not express TMPRSS2 [33].

Here, using genetic depletion of all three dynamin isoforms, we surveyed a range of animal viruses for their dynamin-dependency to infect cells. We show that while some viruses, including SARS-CoV-2, strongly depend on the presence of dynamins to productively infect cells, other animal viruses, including alphaviruses, influenza, and bunyaviruses, among others, can use dynamin-independent endocytosis (DIE) as an alternative and efficient entry mechanism. This yet uncharacterized endocytic pathway appears sensitive to perturbations of the actin cytoskeleton.

## Results and discussion

### Characterization of dynamin 1,2 conditional knock-out cells to study virus entry

Many animal viruses, including the alphaviruses Semliki Forest (SFV) and Sindbis (SINV) virus, influenza virus (IAV), and vesicular stomatitis virus (VSV), have been shown to infect cells using dynamin-dependent endocytosis [2]. Most of these studies have been performed by treating cells with small molecule dynamin inhibitors, by overexpression of dominant-negative inactive forms of dynamin, or by depletion of dynamin mRNAs using RNA interference methods. These loss-of-function approaches, although easy to implement, can suffer from uncharacterized off-targets effects [19,20] or incomplete depletion of the dynamin isoforms.

To address the role of dynamins in virus infection, and to overcome the above-mentioned limitations, we used genetically engineered dynamin 1,2 conditional double knockout (KO) mouse embryonic fibroblasts (MEF$^{DKO}$) [34]. In these cells, the two main isoforms of dynamin (dyn1,2) can be completely depleted within 6 days of cell treatment with 4OH-tamoxifen (4OH-TMX) [34] (Fig 1A). The third isoform of dynamin (dyn3) is not detectable in these cells [34]. To functionally monitor the specificity and level of inhibition of dynamin-dependent endocytosis in this model system, we used transferrin (Tf) as a positive control, which is internalized in a dynamin-dependent manner [35], and the cholera toxin subunit B (CTB),

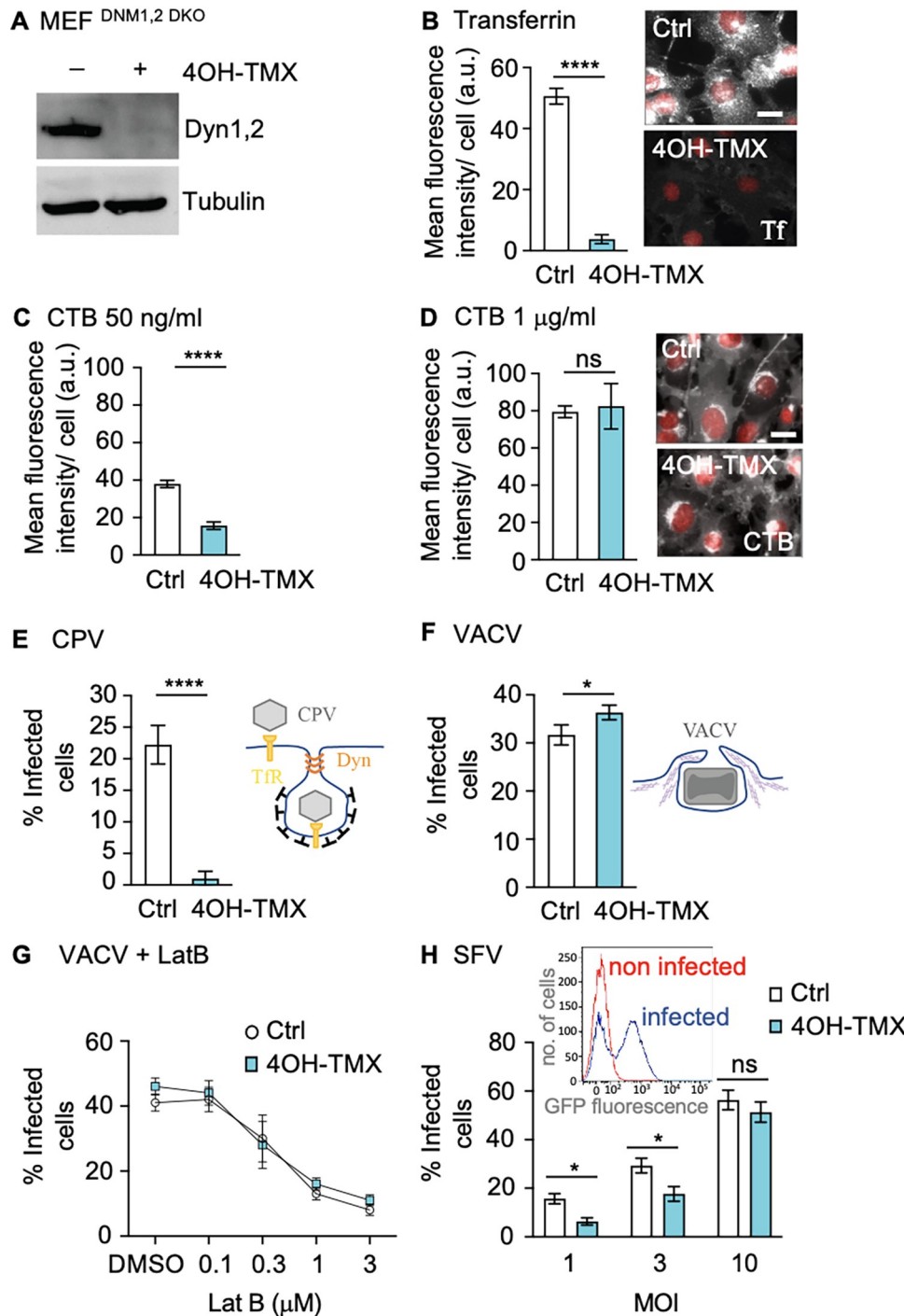

**Fig 1. Characterization of MEF dynamin 1,2 conditional KO cells to study virus infection.** A) Western blot analysis of dynamin 1 and 2 levels in MEF[DKO] cells treated with vehicle control or 4OH-TMX for 6 days. Tubulin was used as a loading control. The Dyn1,2 antibody used recognizes both dynamin 1 and 2. B) Quantification of Tf Alexa-647 (Tf) (5 µg/ml) following a 30 min uptake in MEF[DKO] cells treated with vehicle control or 4OH-TMX for 6 days prior fixation and Hoechst DNA staining. Representative fluorescent images on the right show Tf (white) and Hoechst (red) in MEF[DKO] cells treated with vehicle control or 4OH-TMX. C-D) Quantification of CTB Alexa-647 (CTB) uptake in MEF[DKO] cells treated with vehicle control or 4OH-TMX for 6 days and incubated with indicated concentrations of CTB for 30 min prior to fixation and Hoechst DNA staining. Representative fluorescent images on right show CTB (white) and Hoechst (red) in MEF[DKO] cells treated with vehicle control or 4OH-TMX. E) Quantification of CPV infection in MEF[DKO] cells treated with vehicle control or 4OH-TMX for 6 days and infected with CPV for 24h, and

immunostained for non-structural protein 1 (NS1). The inset schematic illustrates the entry mechanism of CPV. F) Quantification of VACV infection in MEF$^{DKO}$ cells treated with vehicle control or 4OH-TMX for 6 days and infected with VACS for 6 h. The inset schematic illustrates VACV entry by micropinocytosis. G) Effect of Latrunculin-B (LatB) on VACV infection in MEF$^{DKO}$ cells treated with vehicle control or 4OH-TMX for 6 days and infected with VACS for 6 h. H) Quantification of SFV infection in MEF$^{DKO}$ cells treated with vehicle control or 4OH-TMX for 6 days and infected with indicated MOIs of SFV-EGFP for 7 h. The inset illustrates the FACS analysis of virus-induced EGFP fluorescence in non-infected (red line) and infected (blue line) vehicle-control treated cells. Values represent the mean of 3 independent experiments. Error bars represent the standard deviation (STDEV). Statistical significance was calculated using a unpaired two-tailed t-test (*p<0,05; ****p<0,0001; n.s. = non-significant).

which can enter cells via both dynamin-dependent [36,37] (at low doses) or dynamin-independent (at high doses) endocytic mechanisms [38]. All the original data shown in the graphs presented in this work are available in supplementary data file "S1 Data". As expected, 6 days after 4OH-TMX treatment, the uptake of fluorescently labelled Tf (5 μg/ml) was strongly inhibited in most cells (Fig 1B). Consistent with a previous report [34], in all of our experiments, approximately 3–5% of the MEF$^{DKO}$ cells did not respond to 4OH-TMX treatments and as a result maintained normal levels of Tf uptake (S1A–S1C Fig). Importantly, the block in Tf uptake was not due to decreased levels of the Tf receptor at the cell surface, as demonstrated by increased binding of fluorescent Tf at the cell surface of dynamin depleted cells (S1D–S1E Fig). Likewise, the internalization of CTB was also significantly reduced at low toxin concentration (i.e., 50 ng/ml) (Fig 1C), while at higher concentration (i.e., 1 μg/ml), the levels of CTB uptake in MEF$^{DKO}$ cells were comparable to those of vehicle control treated cells (Fig 1D). These experiments confirmed that the MEF$^{DKO}$ represent a suitable model system to study dynamin-dependent and -independent endocytosis.

To test whether the inducible dynamin depletion system was suitable to study virus infection, we first tested three well characterized viruses in the MEF$^{DKO}$ cells treated with vehicle control (EtOH) or 4OH-TMX for 6 days: i) canine parvovirus (CPV), a small single-stranded DNA virus that uses the Tf receptor (TfR) and dynamin-dependent endocytosis to enter cells [39,40] (here used as a positive control); ii) vaccinia virus (VACV-EGFP, EGFP expressed under viral early gene promoter [41]), a large DNA virus that enters cells via actin-dependent, dynamin-independent macropinocytosis [10,11] (negative control); and Semliki Forest virus (SFV-GFP, GFP expressed as a fusion with viral replicase protein nsP3 [42]), which has been shown to infect cells mainly via dynamin-dependent endocytosis [15], although alternative entry mechanisms have also been proposed [16]. Infection rates were determined by fluorescence-activated cell sorting (FACS) flow cytometry analysis of virus-induced expression of GFP (for VACV-GFP and SFV-GFP) or following immunofluorescence staining of viral proteins (for CPV). In the absence of the canine TfR MEF$^{DK}$ cells are not susceptible to CPV infection, indicating that the cell entry process of this virus relies on receptor-mediated endocytosis and no other murine receptors can be efficiently use to facilita virus entry (S2A Fig, non transfected). As expected, transient expression of the feline TfR [40] was sufficient to extablish CPV infection and viral protein synthesis (S2B Fig, Ctrl). 4OH-TMX treatment in MEF$^{DKO}$ cells transiently over-expressing the feline TfR [40] inhibited CPV infection by more than 90% (Figs 1E and S2B). The infectivity of VACV in dyn1,2 depleted MEF$^{DKO}$ cells was comparable to that of control cells, confirming that this virus enters cells via a dynamin-independent mechanism (Fig 1F). Disruption of the actin cytoskeleton using the actin depolymerizing drug Latrunculin-B (LatB), on the other hand, blocked VACV infection in both dyn1,2 depleted and control MEF$^{DKO}$ cells (Fig 1G). The actin-dependency of VACV infection was expected and consistent with viral entry through macropinocytosis, a process where actin polymerization is required to support the formation of PM protrusions that can engulf extracellular fluids and large particles such as VACV virions that have a size of approximately

250x250x350 nm [10,11,43]. Notably, the infection of SFV was only partially inhibited by dyn1,2 depletion, and increasing the virus dose (i.e., the multiplicity of infection, MOI) from MOI 1 to 10 fully restored infection (Fig 1H).

In summary, the MEF$^{DKO}$ cells are a suitable model to study the role of dynamins in virus infection. Small viruses, e.g. CPV, that target a receptor (e.g. TfR) that is internalized exclusively via dynamin-dependent endocytic routes cannot efficiently infect cells in the absence of dynamins. Compensatory dynamin-independent endocytosis is not available for such single-receptor binding viruses. On the other hand, viruses such as SFV, which may target more than one protein receptor, including a variety of heparan-sulphate-containing glycoproteins [44], are internalized primarily via dynamin-dependent endocytosis. If this entry mechanism is not available, infection can occur through an alternative dynamin-independent pathway, albeit less effectively. Regardless of the mechanism of endocytosis, SFV infection in these cells appears to rely on negatively-charged receptors, such as the heparan-sulphate-containing glycoproteins [44], as shown by competition experiments with increasing concentrations of heparin blocking SFV infection in both control and dynamin depleted cells to a similar extent (S2C–S2D Fig).

## Dynamin-independent endocytosis is an alternative, efficient entry pathway for multiple animal viruses

Endocytic pathways that require dynamins, such as clathrin-mediated endocytosis (CME), have been associated with the cell entry of numerous viruses, including members of the mosquito-delivered alphaviruses, such as SFV, SINV [45], and IAV [17], as well as members of *Bunyaviridae* such as Uukuniemi virus (UUKV) [46], vesiculoviruses such as vesicular stomatitis virus (VSV) [47,48], common cold human rhinovirus (RV) B14 and A89 [45,49], and species C AdV such as AdV-C2 or C5 [50–53] as well as AdV-B3 and AdV-B35, although the dynamin requirement of B3 and B35 has been shown to be variable between cell lines [54,55]. Influenza virus has been shown to enter cells also by a micropinocytosis-like mechanism [18]. The results obtained here with SFV (Fig 1H) prompted us to test if the dynamin-independent pathway could be used as a productive entry route for other viruses that are known to use dynamin-dependent endocytosis to enter host cells. To this end, we tested if the infectivity of SINV, VSV, IAV, UUKV, RV-A1 and AdV-C5 was inhibited in dyn1,2 depleted MEF$^{DKO}$ cells. VACV, which is known to infect cells via actin-dependent macropinocytosis irrespective of the presence of dynamins (Fig 1F–1G), was used as a comparison. Infection was monitored by FACS or immunofluorescence analysis. VACV [41], VSV [48], and AdV-C5 [56] were engineered to express the EGFP protein, and SINV to express the mCherry, as fluorescent reporters of infection. MEF$^{DKO}$ cells were treated with vehicle or 4OH-TMX for 6 days, and the infection rates for these recombinant viruses were quantified at 7 hours post infection (hpi) (VACV-EGFP, VSV-EGFP, SINV-mCherry) and 22 hpi (AdV-C5-EGFP) by monitoring the number of EGFP or mCherry expressing cells, or by using immunofluorescence staining against viral antigens with virus-specific antibodies at 8 hpi (IAV X31, UUKV) and 24 hpi (RV-A1). Interestingly, all the tested viruses, albeit with different efficiency, were able to infect cells in the absence of dynamins (Fig 2). At low viral doses (equivalent to an infection of approximately 10–30% of cells), the infections of SINV-mCherry, VSV-EGFP, and IAV X31 were significantly decreased in dyn1,2 depleted cells. In contrast, in the absence of dynamins, the infection of UUKV was not significantly blocked, and in the case of the common cold viruses RV-A1 and AdV-C5-EGFP, infection in dynamin-depleted cells was enhanced in comparison to controls (Figs 2 and S3). Thus, dynamin-independent endocytosis is an efficient, alternative virus entry portal that can be exploited by multiple animal viruses. The results also

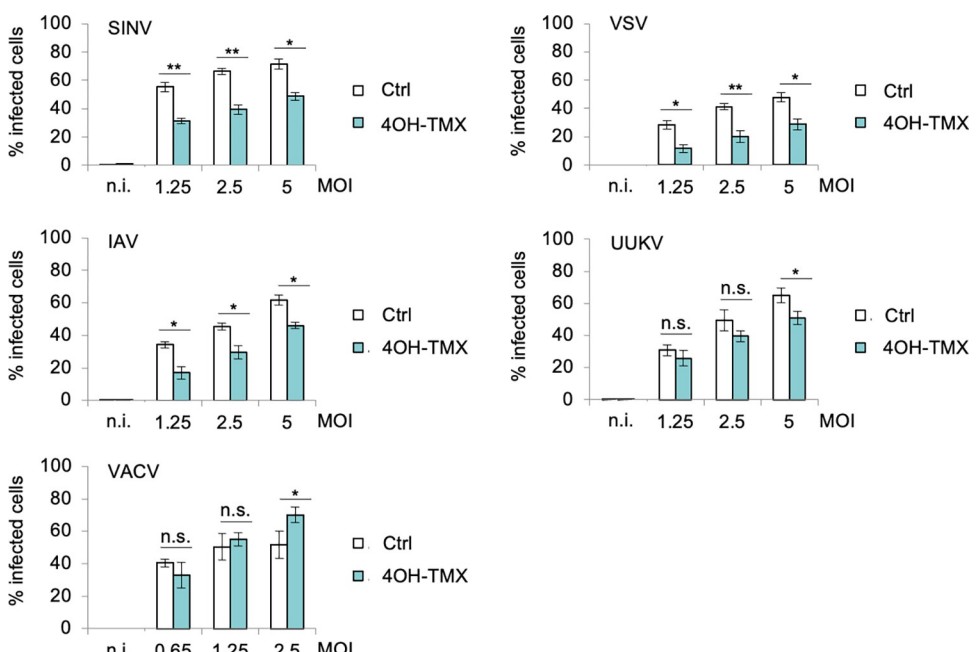

**Fig 2. Dynamin-independent endocytosis is an alternative, efficient virus entry pathway for multiple animal viruses.** Infection of indicated viruses in MEF[DKO] cells treated with vehicle control or 4OH-TMX for 6 days and infected for 6 h (VACV), 7 h (SINV, VSV), and 8 h (IAV X31, UUKV). Virus infection was determined by FACS analysis of EGFP (VAVC and VSV), mCherry (SINV), or after immunofluorescence of viral antigens using virus-specific antibodies (IAV X31 and UUKV). Values indicate the mean of three independent experiments and the error bars represent the standard deviation (STDEV). n.i. = non infected. Statistical significance was calculated by unpaired two-tailed t-test (*p<0,05; ** p<0,01; n.s. = non-significant).

demonstrate that using a low viral dose is advisable to estimate the contribution of the two entry mechanisms to infection. A comparative analysis of all tested viruses, at a viral dose that corresponds to 20–40% infection rates in vehicle control treated cells (Figs 2 and S2), indicates that the dependence on dynamin-mediated endocytosis follows this qualitative order: VSV>SFV = SINV = IAV>UUNV. Infections with VACV, RV-A1 and AdV-C5 were not inhibited by dynamin depletion, rather increased, suggesting that the cells may have adapted to the depletion of dynamins by up-regulation of dynamin-independent endocytic pathways. Upond complete depletion of dynamins, MEF[DKO] cells stop dividing due to a block in cyto-dieresis, a process that requires dynamin function [34]. However, the infection observed by multiple viruses in dynamin-depleted cells demonstrates that the lack of cell division does not prevent the virus entry process and expression of viral genes, at least of the viruses used in this study.

## Dynamin-independent virus entry is facilitated by the actin cytoskeleton

Dynamin-depleted cells were efficiently infected by SFV at moderate virus doses (Fig 1H). A set of small molecule inhibitors, known to block different endocytic pathways, was used to further characterize the dynamin-independent endocytic mechanism used by SFV. Cells, pre-treated for 6 days with vehicle control or 4OH-TMX, received the drugs 15 min before infection and the drugs were present during virus inoculation for 30 min. To limit potential toxic effects, drugs and virus inoculum were removed after 30 min. To prevent further SFV cytosolic entry following removal of virus inoculum and drugs, cells were maintained in medium containing NH4Cl, which neutralized endosomal pH (Fig 3A). The number of infected cells was

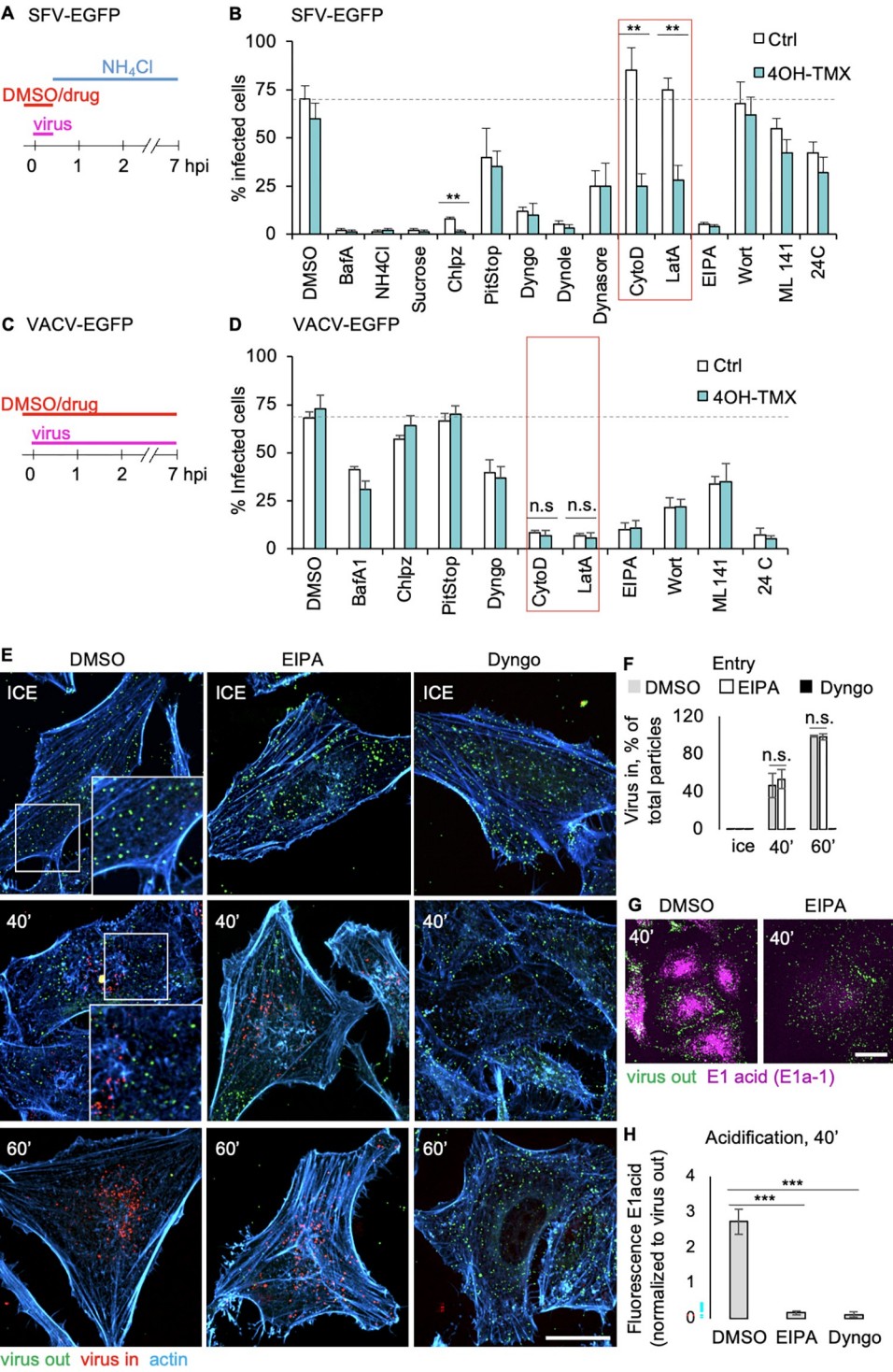

**Fig 3. Dynamin-independent virus entry is actin dependent.** A) Schematic description of indicated drug treatments in MEF[DKO] cells following 6-day treatment with vehicle (ctrl) or 4OH-TMX, and infected with SFV-EGFP. B) Quantification of the experiment described in A. After drug treatment and infection, the percentage of infected cells was determined by FACS analysis of virus-induced expression of EGFP. The red-boxed area indicates the treatments with actin depolymerizing drugs. C) Schematic description of indicated drug treatments in MEF[DKO] cells following 6-day treatment with vehicle (ctrl) or 4OH-TMX, and infected with VACV-EGFP. D) Quantification of the experiment described in C. After drug treatment and infection, the percentage of infected cells was determined by FACS analysis of virus induced expression of EGFP. The red-boxed area indicates the treatments with actin

depolymerizing drugs. Values indicate the mean of three independent experiments and the error bars represent the standard deviation. Statistical significance was calculated by unpaired two-tailed t-test (** p<0,01; n.s. = non-significant). E) Representative confocal images of virus entry in MEF$^{TKO}$cells pretreated with 4OH-TMX for 6 days, incubated with DMSO control, EIPA (80 µM), or Dyngo-4A (25 µM) and viruses (equivalent MOI = 100) 30 minutes before incubation on ice for 120 min. After washes, cells where incubated with DMSO or drugs at 37˚C and fixed at indicated times before processing for sequential immunofluorescence as described in S4C Fig. Inset images show a magnification of the area indicated by the white boxes. Virus out = non internalized viruses; virus in = internalized viruses. F) Quantification of non-internalized (virus out) and internalized (virus in) virions using automated image analysis. Values represent the mean of >5 cells from 3 independent experiments, and error bars represent the STDEV. Scale bar = 10 µm. Statistical analysis was performed using ordinary two-way ANOVA multiple comparison test (n.s. = non significant; *** p<0.001). G) Representative confocal images of virus entry and acidification of the E1 envelope protein by sequential immunofluorescence as described in E using a monoclonal antibody against the acid form of the E1 glycoprotein (E1 acid) and a combination of antibodies agains E1 and e2 to detect virions at the outer side of the plasma membrane (virus out). Scale bar = 10 µm. H). Quantification of non-internalized (virus out) and internalized virues detected by the E1a-1 antibody against the acid form of SFV E1 glycoprotein using image analysis performed with the Imaris software. Values represent the mean of >5 cells from 3 independent experiments, and error bars represent the STDEV. Scale bar = 10 µm. Statistical analysis was performed using ordinary two-way ANOVA multiple comparison test (n.s. = non significant; *** p<0.001).

then assessed at 7 hpi. The concentrations and cellular target of the selected drugs used in this experiment are listed in S4A Fig. Both in control and dyn1,2 depleted cells, SFV infection was highly sensitive to hyper-osmotic shock induced by 0.45 M sucrose, a treatment that blocks endocytosis, and drugs known to inhibit endosomal acidification, i.e. Bafilomycin A1 (BafA) and NH$_4$Cl (Fig 3B). Thus, even in the absence of dynamin, SFV infection required endocytosis and, as described earlier, delivery of the viral particles to acidic intracellular endosomes [15]. Unexpectedly, inhibitors of CME (e.g. chlorpromazine and PitStop) [57], and dynamins [58] (i.e. Dyngo-4A and Dynole [59]) blocked infection to a similar extent in both control cells as well as dyn1,2 depleted MEF$^{DKO}$ cells, indicating off-target effects of these drugs.

Incubation of cells with the phosphoinhositol 3-kinase (PI3K) inhibitor wortmannin, Rac1/CdC42 GTPase inhibitor ML141, or at lower temperatures (i.e. 24˚C), the three treatments that are known to specifically block macropinocytosis, were either ineffective (wortmannin) or moderately effective (ML142 and low temperature) to a similar extent in both control and dyn1,2-depleted MEF$^{DKO}$ cells, indicating that these endocytic processes are different from *bona fide* macropinocytosis. Indeed, the same treatments significantly inhibited VACV infection (Fig 3C and 3D).

Treatments with actin depolymerizing agents such as cytochalasin-D (CytoD) and Lat-A, did not inhibit SFV infection in control MEF$^{DKO}$ cells (Fig 3B, Ctrl bars). Thus, unlike the large VACV (Fig 3D, CytD and LatA), the smaller SFV (approximately 60–70 nm in diameter) [60] does not require functional actin cytoskeleton for the entry process in genetically unperturbed cells. However, in dyn1,2-deficient cells, the treatment with CytoD and Lat-A significantly reduced SFV infection by approximately 60% compared to control cells treated with the same drugs (Fig 3B, red boxed area). Hence, dynamin-dependent and -independent endocytic pathways differ in their sensitivity to actin-depolymerizing drugs, even for a small virus.

The amiloride EIPA, an inhibitor of the sodium-hidrogen antiporter, often used to block micropinocytosis also inhibited SFV infection even in WT MEF cells. Is is know that this inhibitor also causes neutralization of endosomal pH [61], which could explain the unexpected result if after EIPA treatment the viral spike protins failed to undergo the low pH-dependent conformational changes required to activate virus fusion with endosomal membranes. To address the impact of EIPA on SFV entry and envelope protein acidification, we perdormed direct virus entry and E1 protein acid-conformation changes assays in dynamin depleted MEF cells. To this end, cells there preincubated with vehicle DMSO, EIPA (80 µM), of Dyngo-4A (25 µM, here used as a positive control for inhibition of virus entry) for 30 min before shifting

the cultures on ice and incubation with SFV (equivalent MOI = 100) for 2 hours. After removing the unbound virus, the cultures were shifted to 37°C for indicated times in the presence of replenished DMSO control, EIPA, or Dyngo-4A, and 50 μM cycloheximide to prevent viral protein synthesis. After fixation, cells were processed for sequential immunofluorescence staining of the viral envelope proteins E1 and E2 to distinguish viral particles outside of the PM, i.e. particles not yet internalized (virus out), from internalized virions (virus in) (S4C–S4D Fig). In this technique, non-internalized viruses are immunostained in fixed cells before permeabilization, using a combination of polyclonal antibody against the viral E1 and E2 glycoproteins, followed by secondary antibodies conjugated to a fluorophore (S4C Fig, Ab 1, magenta). After a second fixation, cells are permeabilized and immunostained with the same E1-E2 antibodies followed by secondary antibodies conjugated to a different fluorophore (S4C Fig, Ab 2, green). Hence, in this assay, non-internalized virions are stained with the first fluorophore (Fig 3E, green spots, virus out). Internalized virions are stained only with the second fluorophore (Fig 3E, red dots, virus in). As shown in Fig 3E and 3F, EIPA did not inhibit SFV entry while Dyngo-4A blocked all virions on the outer surface of the PM even after 60 min uncubation at 37°C.

To asses a possible detrimental effect of EIPA on E1 acid-induced conformation change that occurs in endosomes, which could explain the inhibition of infection observed in both WT and dynamin-depleted cells (Fig 3B), a combination of E1-E2 antibodies (to recognize viruses outside the PM) and a monoclonal antibody (E1a-1, viruses inside the cells) that recognizes the acid conformation of E1 (Fig 3G and 3H) [62] was used in sequential immunostaining assays as described above. The results showed that while EIPA does not inhibit SFV entry (Fig 3E and 3F, DMSO and EIPA, and S4C Fig), it rather prevents the conversion of the E1 protein on internalized virions from neutral to acid-induced conformation (Fig 3G and 3H), presumably due to the reported neutralization of endosomal pH [61] at the concentrations used to effectively inhibit micropinocytosis. The dynamin inhibitor Dyngo-4A, on the other hand, strongly inhibited SFV entry even if the cells were devoid of dynamins (Fig 3E and 3F, Dyngo).

The lack of infection inhibition with the remaining micropinocytosis inhibitors, which otherwise did inhibit VACV infection, excludes a role for macropinocytosis in the SFV infection of both control and dyn1,2 depleted cells.

Cell viability assay confirmed that under the experimental conditions used in this study, none of the drugs that bloked SFV virus infection significantly reduced cell viability (S4B Fig).

## Ultrastructural analysis of dynamin-independent SFV entry

Because SFV entered cells efficiently both in the presence and absence of dynamins, we used transmission electron microscopy (TEM) to gain ultrastructural information on the endocytic processes that mediate SFV entry in dyn1,2 depleted cells. MEF$^{DKO}$ cells were treated with 4OH-TMX or vehicle control for 6 days and, following virus (MOI = 1000) adsorption at 4°C for 1 h, cells were shifted to 37°C for 15 minutes to promote viral internalization. In MEF$^{DKO}$ cells treated with vehicle control, TEM analysis revealed numerous viruses at the outer surface of the cells (Fig 4A), as well as inside endocytic invaginations that were surrounded by an electron dense coat, consistent with the appearance of clathrin coated pits [63,64] (CCP) (Fig 4B). A sizable fraction of SFV particles were also found inside bulb-shaped non-coated pits (NCP) (Fig 4C). In addition to small PM invaginations, large (>250 nm in diameter) vacuoles containing viruses and located close to the PM were detected. These structures, here annotated as 'large endocytic/endosomal profiles' (LEP), could represent early endosomal vesicles or large invaginations of the PM that appear circular in TEM cross-sections (Fig 4D, Ctrl). In dyn1,2

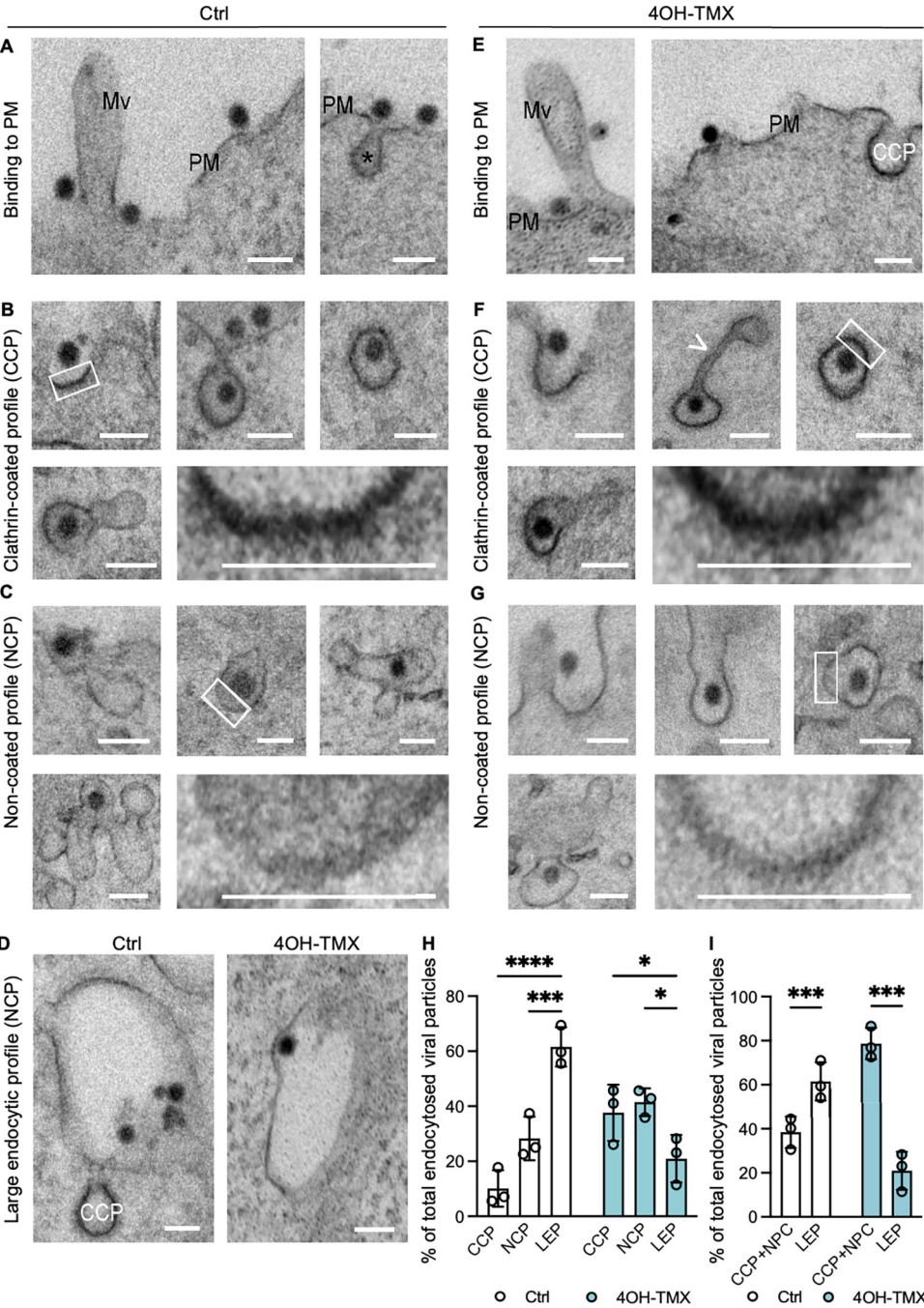

**Fig 4. Ultrastructural analysis of Semliki Forest Virus entry in MEF$^{DKO}$ cells.** Representative TEM images of MEF$^{DKO}$ cells treated with vehicle control (A-D) or 4OH-TMX (E-G, and D, 4OH-TMX) for 6 days and infected with SFV (MOI 1000) on ice for 2 h followed by shift to 37°C for 15 min before fixation and processing for TEM. The fractions of total viral particles found in each of the described endocytic processes are quantified in H and I. Scale bar 100 nm. CCP = clathrin-coated profile, Mv = microvilli, PM = plasma membrane. The asterisks (*) in panel A indicate an endocytic process. White arrowheads indicate elongated tubular membranous structures connected to CCP. Boxed areas are magnified at bottom right corner of each figure panel. All values represent the mean and standard deviation of three replicas. Quantification of each treatment (EtOH vehicle ctrl or 4OH-TMX) includes over 120 viral particles. Statistical analysis was performed using ordinary two-way ANOVA multiple comparisons test (* $p<0.05$; *** $p<0.001$; **** $p<0.0001$). Scale bars = 5 μm.

depleted MEF$^{DKO}$ cells, SFV virions were also readily detected on the PM (Fig 4E). It has been previously reported that in the absence of dynamins the final step of CME does not occur, and the respective 'stalled' vesicles are pulled from the PM towards the cytoplasm by actin polymerisation followed by depolymerization and release back towards the PM [34]. In agreement with these reports [34], in dyn1,2 depleted MEF$^{DKO}$ cells, CCPs were often associated with long tubular membraneous structures (Fig 4F, white arrowhead). A fraction of the viruses was found in these stalled CCPs (Fig 4F), and the rest in NCPs (Fig 4G) and in LEPs (Fig 4D, 4OH-TMX). Consistent with the fast entry kinetics of SFV, quantitative analysis of the TEM images revealed that 10 min after entry in Ctrl cells, 60% of the virions were localized in LEP, which included endosomes, 10% in CCP, and approximately 30% in NCP (Fig 4H, Ctrl). Upon dynamin depletion, this ratio changed (Fig 4H, 4OH-TMX), and the number of virions in CCPs and NPCs at the PM increased at the expense of viruses found in the endosomal proceses. Cumulative analysis revealed that in the absence of dynains up to 70–90% of viral particles where in CCP and NCP and only 10–30% in large endocytic structures, indicating a strong inhibition of virus endocytosis (Fig 4I).

This analysis is consistent with two entry mechanisms for SFV, one dynamin-dependent and another dynamin-independent. We speculate that upon depletion of dynamins, a fraction of the virus remains trapped in CCPs and NCPs. This stalled entry pathways might account for the partial inhibition of infection in dyn1,2 depleted MEF$^{DKO}$ cells observed in this study for SFV and possibly for other viruses. The fact that in unperturbed cells, SFV infection is not sensitive to actin depolymerizing drugs, suggests that the dynamin-dependent route is the predominant entry pathway. However, a sizable fraction of the virions can access an alternative entry pathway that, based on the effect of actin depolymerizing drugs, seems to rely more on the actin cytoskeleton. The precise role of actin is unknown but it could potentially help the final pinching step of the dynamin-independent endocytic process.

## ACE2-mediated endocytosis of SARS-CoV-2 trimeric spike is mainly dynamin-dependent

The exact mechanism of SARS-CoV-2 cell entry is not fully understood, and evidence suggests that both clathrin-dependent and -independent endocytosis could be involved in cell culture models, including human neurons that do not express the cell surface protease TMPRSS2 [33]. A study from Bayati et al. [65] used chemical inhibitors of dynamins and siRNA-mediated depletion of clathrin to show that the internalization of both the SARS-CoV-2 trimeric soluble Spike (S) protein and the infection of lentiviruses pseudotyped with SARS-CoV-2 S were decreased. Another study indicated that cell entry of SARS-CoV-2 Spike protein occurs via clathrin independent endocytosis in cells devoid of the human angiotensin converting enzyme 2 (ACE2) [66]. Both studies implied entry via dynamin-dependent mechanisms.

To address the role of endocytosis in SARS-CoV-2 infection, and to identify the potential endocytic mechanisms that leads to productive entry, we started our investigation by following the uptake and intracellular trafficking of fluorescently labelled, recombinant soluble trimeric SARS-CoV-2S protein. This soluble version of the viral S protein is held as a trimer by a molecular clamp [67,68] that replaces the trans-membrane domain of the glycoprotein. In addition, for stabilization purposes, the polybasic furin-cleavage site, required to activate the viral spike for fusion, was rendered uncleavable in each monomer by mutagenesis [49]. This likely also blinded the S-trimer to neuropilin-1 (NRP1) binding [69]. MEF$^{DKO}$ cells transiently expressing the main viral receptor ACE2 [70], and an EGFP-tagged version of the early endosome protein Rab5 [71], were used in these studies. Time-course uptake assays followed by confocal fluorescence microscopy were performed to distinguish the fraction of Alexa Fluor-

555-labelled S localized at the PM from those internalized into endosomal vesicles labelled by Rab5-EGFP (early endosomes) (Fig 5). Tf labelled with Alexa Fluor-647 was used as an internal positive control to monitor the extent of inhibition of dynamin-dependent endocytosis in each cell analysed.

In the absence of ACE2, MEF cells did not significantky bind SARS-COV-2 S (Fig 5A and 5B). In vehicle control treated MEF<sup>DKO</sup> cells transiently expressing ACE2 and Rab5-EGFP, the uptake of S was efficient, and the internalized S protein colocalized with Rab5-EGFP positive vesicles (Fig 5C and 5D, Ctrl). In dyn1,2 depleted MEF<sup>DKO</sup> cells most of the S fluorescent signal remained associated with the PM of the cells, even after 3h, indicating a strong, albeit not complete, inhibition of endocytosis (Fig 5C and 5D, 4OH-TMX). Consequently, the colocalization of the intracellular Rab5 compartment with the S protein remained low throughout the time course (Fig 5A and 5B, 4OH-TMX). Similar results were obtained for fluorescently labelled Tf, which also accumulated on the PM in dyn1,2 depleted MEF<sup>DKO</sup> cells at the expenses of intracellular compartments (Fig 5A, 4OH-TMX). Taken together, these results demonstrate that, similarly to Tf, the ACE2-mediated endocytosis of SARS-CoV-2 S trimeric proteins is mainly dynamin-dependent.

## Depletion of dynamins blocks endocytosis of authentic SARS-CoV-2 virions

In addition to the MEF<sup>DKO</sup> cells, a triple dynamin 1,2,3 conditional KO cell line (here referred to as MEF<sup>TKO</sup>) [72] became available for this study. The dyn1,2,3-depletion in MEF<sup>TKO</sup> cells following a 6-day treatment with 4OH-TMX, blocks Tf uptake similarly to the dyn1,2-depletion in MEF<sup>DKO</sup> cells [72]. An advantage of this triple dyn1,2,3 KO model system is that in addition to depletion of all three dynamin isoforms 1, 2, and 3, these cells adhere better to culture surfaces compared to the MEF<sup>DKO</sup> cell clone. In addition, the responsiveness to 4OH-TMX treatment is more stable over cell passaging than in MEF<sup>DKO</sup> cells (not shown). The dynamin inhibitor OH-dynasore (also called Dingo-4A) showed off-target effects in both MEF<sup>DKO</sup> and MEF<sup>TKO</sup> cells, bloking the infection of the alphavirus SFV in control and dynamin-depleted cells to a similar extent, when tested both at high and moderate drug concentrations (S5A and S5B Fig).

Because murine cells are not infectable by SARS-CoV-2, to study how a clinical isolate of the ancestral SARS-CoV-2 Wuhan strain (here referred to as Wuhan [73]) enters MEF<sup>TKO</sup> cells, we used a lentiviral expression vector to stably express the human ACE2 receptor (MEF<sup>TKO</sup>-ACE2). To confirm that the inhibition of S internalization observed in dynamin depleted cells corresponded to a proportional block in endocytosis of SARS-CoV-2 virions, we directly measured the extent of virus internalization in dyn1,2,3-depleted and control MEF<sup>TKO</sup>-ACE2 cells. Viruses (Wuhan, equivalent MOI of 10) were added to cells at 37°C for 60 min in the presence of 50 μM cycloheximide to prevent viral protein synthesis. After fixation, cells were processed for sequential immunofluorescence staining as described above for SFV to distinguish viral particles outside of the PM, i.e. particles not yet internalized (virus out), from internalized virions (virus in). Hence, in this assay, non-internalized virions are stained either with the first (i.e. magenta spots) or with both (i.e. white colocalizing spots) fluorophores (Fig 6A, Ctrl, magenta or white dots). Internalized virions are stained only with the second fluorophore (Fig 6A, 4OH-TMX, green dots).

In the unperturbed control cells, at 45 min p.i., image analysis after confocal imaging revealed that a sizable fraction of viruses was found inside the cells (Fig 6B, Ctrl). In contrast, depletion of dyn1,2,3 blocked virus endocytosis almost completely (Fig 6B, 4OH-TMX). This reduction of virus internalization did not correlate with proportional changes in the cell

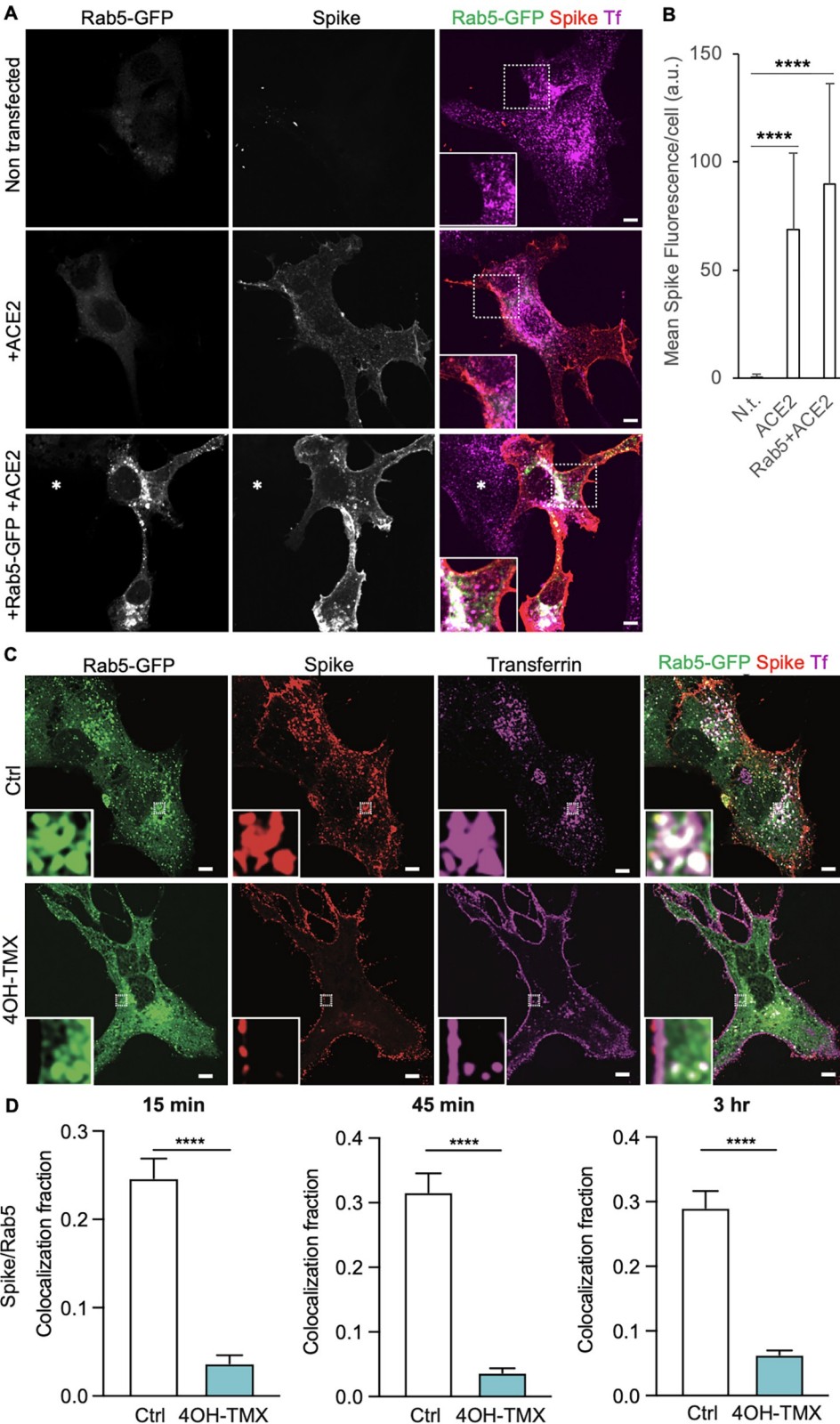

**Fig 5. ACE2-mediated endocytosis of SARS-CoV-2 trimeric spike proteins is dynamin-dependent.** A) Representative confocal fluorescence images of MEF$^{DKO}$ cells non transfected, transiently expressing hACE2, or hACE2 and Rab5-EGFP. B) Quantification of the mean spike fluorescence per cell. A.u. = arbitrary units. C)

Representative confocal fluorescence images of MEF$^{DKO}$ cells transiently expressing hACE2 and Rab5-EGFP and treated with vehicle control or 4OH-TMX for 6 days and incubated for 3h with Alexa Fluor-555-labelled transferrin (Tf, magenta) and Alexa Fluor-647 labeled soluble trimeric SARS-CoV-2 Spike protein (red). The insets in A) and C) show magnified images from the indicated white boxed areas. The images represent a single optical slice of the imaged cell. B) Quantification of colocalization between indicated proteins in cells treated as described in C). The mean ±STDEV of 16 Ctrl cells and 14 4OH-TMX treated cells for 15 min; 16 Ctrl cells and 15 4OH-TMX cells for 45 min; and 17 Ctrl cells and 18 4OH-TMX cells for 3 h are shown. The statistical significance was calculated using a non parametric Mann-Whitney U test (****$P<0,0001$).

surface levels of the receptor ACE2, which decreased less than 20% upon dynamin depletion (S5C and S5D Fig). This experiment demonstrates that the strong block of S internalization observed in dynamin depleted cells corresponds to a similar block in virus endocytosis.

## SARS-CoV-2 infection in ACE2-expressing MEF cells requires endosome maturation and is insensitive to TMPRSS2 inhibitors

To study how a clinical isolate of the ancestral SARS-CoV-2 Wuhan strain [73] infects MEF$^{TKO}$ cells, in addition to the MEF$^{TKO}$-ACE2 cells, we generated a cell line stably expressing ACE2 together with a N-terminally GFP-tagged version of TMPRSS2 (here referred to as MEF$^{TKO}$-AT). We used fluorescence activated cell sorting to isolate cells that expressed high, medium, and low levels of TMPRSS2-GFP. However, only the cells that expressed the lowels levels of GFP reattached to the culture flasks (se materials and methods), indicating that in MEF cells a high surface expression of the trypsin-like serine protease TMPRSS2 interferes with cell adhesion.

We first determined to which extent in these cell lines the cell entry of SARS-CoV-2 Wuhan depended on endosomal proteases or cell surface serine proteases such as TMPRSS2. To this end, we tested the sensitivity of Wuhan infection to nafamostat, an inhibitor of TMPRSS2 [74], and apilimod, an inhibitor of the phosphoinositol-5 kinase (PIP5K) required for efficient early to late endosome maturation and, therefore, delivery of endocytosed cargo to the lysosomal compartment [75,76]. Drug or vehicle-control pretreated infected cells were fixed at 20 hpi and the percentage of Wuhan infection was monitored by immunofluorescence using antibodies against the viral nucleoprotein, followed by automated high-content imaging and image analysis (S5E Fig). As expected, in cells that did not express TMPRSS2, Wuhan infection was strongly inhibited by apilimod but not by Nafamostat (S5E and S5F Fig, MEF$^{TKO}$-ACE2). The over-expression of TMPRSS2-GFP, even if at low levels, had two main effects, firstly, it slightly increased the overall infectivity of the virus in DMSO-control treated cells compared to the values obtained in cells that only expressed ACE2 (S5E and S5G Fig, MEF$^{TKO}$-AT). Secondly, it rendered infection partially resistant to apilimod and sensitive to nafamostat (S5E and S5F Fig, MEF$^{TKO}$-AT). These results indicate that the MEF-ACE2 cells either do not express TMPRSS2 or, if they do, the protein is not available to the virus. Hence, infection depends on virus endocytosis and delivery to endo/lysososmes. In MEF$^{TKO}$-AT, the low levels of TMPRSS2 allow the virus to bypass the need for endosome maturation.

## SARS-CoV-2 uses dynamin-dependent and -independent entry to infect ACE2-expressing MEF cells and both pathways are actin-dependent

To confirm that the dynamin-dependent endocytosis of the S protein and SARS-CoV-2 virions reflected infectious entry pathway of the virus, infection assays were first performed in vehicle control and 4OH-TMX treated MEF$^{TKO}$-ACE2 cells. Two clinical isolates of SARS-CoV-2 were tested, the ancestral Wuhan [73] virus and the more infectious Delta variant [77] (here referred to as Delta). In these experiments, we also investigated the role of the actin

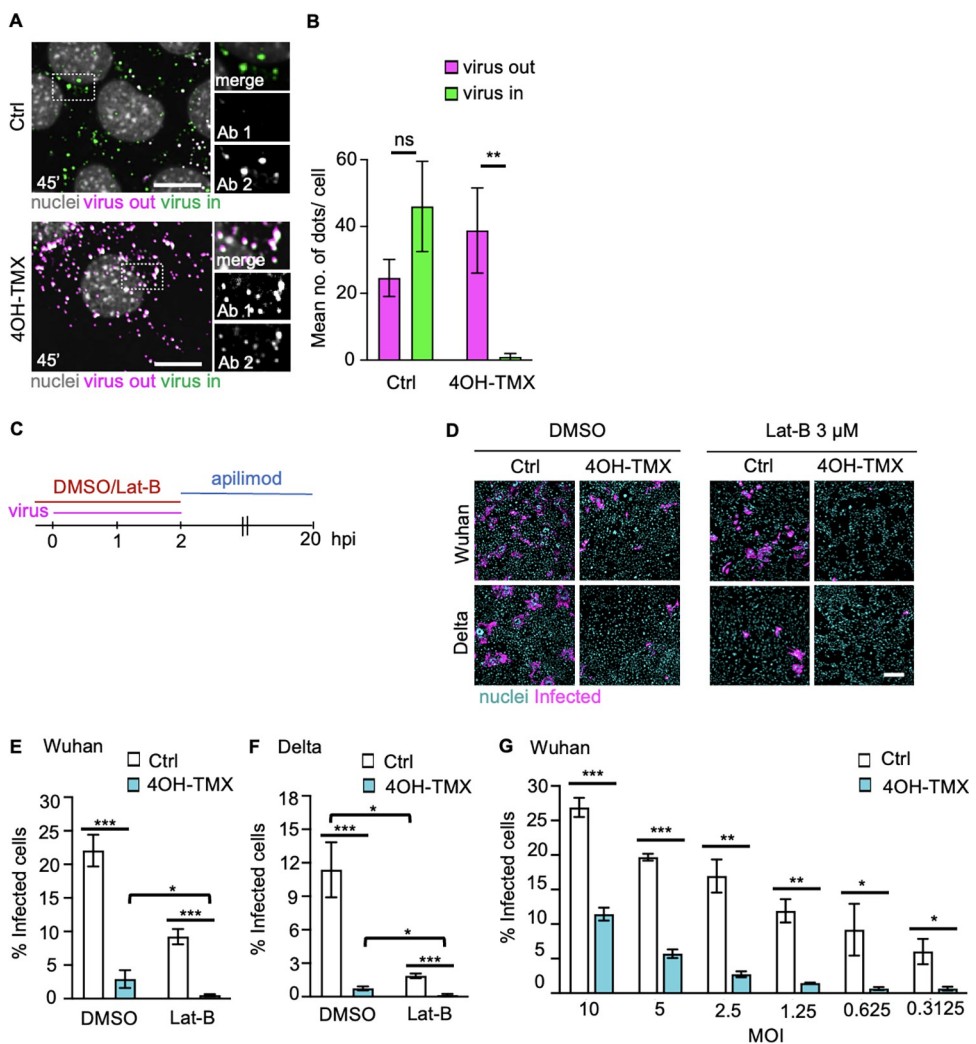

**Fig 6. Dynamin-dependent and -independent entry of SARS-CoV-2 Wuh and Delta infection in ACE2-expressing MEF$^{TKO}$ cells is actin dependent.** B) Representative confocal images of virus entry in MEF$^{TKO}$ACE2 cells (4OH-TMX) or control cells (Ctrl) fixed at 60 min after virus inoculation and processed for sequential immunofluorescence as described in A. Inset images show a magnification of the area indicated by the white dashed boxes, with merged as well as separated fluorescence images. Virus out = non internalized viruses; virus in = internalized viruses. C) Quantification of non-internalized (virus out) and internalized (virus in) virions using automated image analysis. Values represent the mean of 15 cells from 3 independent experiments, and error bars represent the STDEV. Scale bars = 10 μm Statistical significance was calculated by unpaired two-tailed t-test (**p<0,01; n.s. = non-significant). D) Schematic description of the Lat-B treatment in MEF$^{TKO}$ACE2. E) Representative fluorescence images of MEF$^{TKO}$ACE2 treated with indicated compounds 15 minutes before infection and infected with Wuhan or the Delta variant of SARS-CoV-2 for 20 h at 6 days after vehicle control (Ctrl) or 4OH-TMX treatment. Scale bars = 200 μm F-H) Quantification by image analysis of SARS-COV-2 Wuhan or Delta infection in MEF$^{TKO}$ACE2 cells treated with vehicle control (Ctrl) or 4OH-TMX for 6 days and infected with indicated MOIs for 20 h. Values indicate the mean of at least three independent experiments and the error bars represent the STDEV. Statistical significance was calculated using a non parametric Mann-Whitney U test (*p<0,05; **p<0,01; ***p<0,001).

cytoskeleton using the actin depolymerising drug Lat-B. Because prolonged treatments with this drug result in loss of fibroblasts cell morphology and lead to cell detachment, we implemented a procedure to interfere with actin dynamics only during virus entry, without compromising cell attachment (Fig 6C). Six days after 4OH-TMX or vehicle control treatments, MEF$^{TKO-}$ACE2 cells were treated with 3 μg/ml Lat-B for 15 min before infection. Lat-B

was also present in the virus inoculum for 2 additional hours. The medium containing unbound virus and Lat-B was then removed and replenished with new medium containing 2 μM apilimod, to limit further infection after Lat-B removal (Fig 6C). The fraction of infected cells was determined by immunofluorescence imaging and image analysis at 20 hpi. In cells treated with 4OH-TMX for 6 days and DMSO, infection with Wuhan or Delta variants was reduced by up to 80% (Fig 6D–6F, DMSO), indicating that in these cells, dynamin-dependent endocytosis represents the main productive viral entry route. Actin depolymerization inhibited infection by more than 60% in control cells and this inhibitory effect was even stronger in dynamin depleted cells for both tested viruses (Fig 6D-6F, Lat-B). Thus, in ACE2-expressing MEF$^{TKO}$ cells, where TMPRSS2 is ether not expressed or not accessible to the virus, the infection of SARS-CoV-2 is mainly dynamin-dependent, and it is facilitated by the actin cytoskeleton. A similar dynamin- and actin-dependent entry mechanism has been described for other viruses, such as VSV [47], that have a similar size to coronaviruses (i.e. 100–120 nm in diameter including the spikes), and may not completely fit into dynamin-accessible endocytic invaginations. Actin polymerization facilitates the maturation of the endocytic cup, allowing the formation of the narrow membranous neck where dynamin binds and cleaves off the nascent vesicle [47]. Interestingly, similarly to other tested viruses but to a lower extent, increasing the virus load of SARS-CoV-2 Wuhan to an amount sufficient to infect 25% of the cells restored infection up to 12% in a virus-dose dependent manner (Fig 6G). Considering that 3–5% of these cells are not responsive to 4OH-TMX-induced depletion of dynamins, the remaining infection observed in dyn1,2,3-depleted cells indicated virus infection via dynamin-independent endocytosis. Thus, albeit with much lower efficiency compared to the dynamin-dependent entry, at high doses SARS-CoV-2 can also enter cells by an alternative endocytic mechanism.

## Ultrastructural analysis of SARS-CoV-2 virus entry

Ultrastructural TEM analysis of SARS-CoV-2 Wuhan entry in MEF$^{TKO-}$ACE2 was performed by quantitative transmission electron microscopy (Fig 7). Viral particles were first allowed to bind at the cell surface for two hours on ice, at an equivalent MOI of 100. Cells were then shifted at 37°C for 15 minutes to allow endocytosis before fixation and TEM processing. Virions were readily identified at the cell surface of both vehicle control and 4OH-TMX pretreated cells (Fig 7A and 7E). Occasionally, electrondense filaments, with lengths consistent with the 20-nm size of the viral spikes, were observed connecting the virions and the PM (Fig 7A and 7E, white arrow heads, and inset in A). Similar to SFV, in vehicle- and 4OH-TMX-treated cells SARS-CoV-2 viruses were found distributed between CCPs and NCPs, as well as in large endosomal vesicles (Fig 7B, 7C, 7F, 7G and 7D). Quantitative analysis confirmed that upon dynamin depletion the number of viral particles inside CCPs and NCPs increased to more than 80% of the total endocytosed viruses, at the expenses of the virions found inside large endosomal structures (Fig 7H and 7I). Confirming our previous results, this analysis indicated that dynamin-dependent endocytosis is the main endocytic pathway for SARS-CoV-2.

## TMPRSS2 expression bypasses the need of SARS-CoV-2 ancestral strain but not omicron variants for dynamin-dependent endocytosis and endosome maturation

As opposed to small-molecule inhibitors, RNAi or dominant-negative approaches, which can result in uncharacterized off-target effects or incomplete inactivation or depletion of dynamins, the results presented here were obtained in cells where all three isoforms of dynamins were completely depleted by a chemically induced KO. Although coronaviruses are potentially

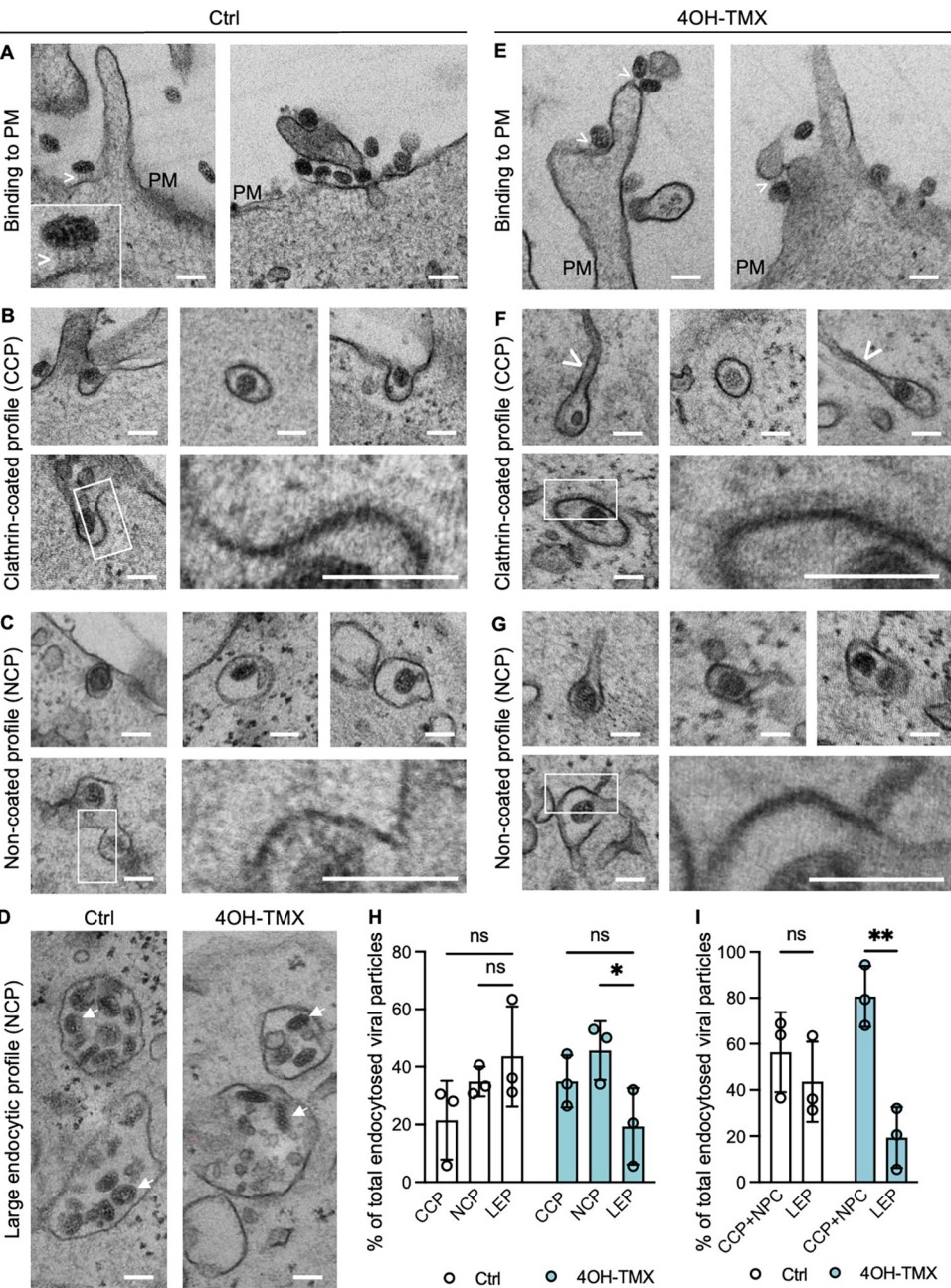

**Fig 7. Ultrastructural analysis of SARS-CoV-2 entry in MEF$^{TKO}$ACE2 cells.** Representative TEM images of MEF$^{TKO}$ACE2 cells treated with vehicle control (A-D) or 4OH-TMX (E-G, and D, 4OH-TMX) for 6 days and infected with SARS-CoV-2 Wuhan (MOI 100) on ice for 2 h followed by shift to 37°C for 15 min before fixation and processing for TEM. The fraction of total viral particles found in each of the described endocytic processes is quantified in H and I. Scale bar 100 nm. CCP = clathrin-coated profile, Mv = microvilli, PM = plasma membrane, asterisks (*) in panel A indicate an endocytic process. White arrowheads in panel A and E indicate electrondense filaments connecting the virions to the cell membrane. Boxed areas are magnified at bottom right corner of each figure panel. All values represent the mean and standard deviation of three replicas. Quantification of each treatment (EtOH vehicle ctrl or 4OH-TMX) includes 120 viral particles per condition. Statistical analysis was performed using ordinary two-way ANOVA multiple comparisons test (* $p < 0.05$; ** $p < 0.01$; ns = non significant).

able to infect cells by direct fusion with the PM, the extent of endocytosis- vs PM-fusion, in different cell types *in vitro* is not known. To determine whether low levels of TMPRSS2 could rescue viral infection in dynamin-depleted cells, infection assays were repeated for SARS-CoV-2 Wuhan, Omicron B1.4, and the more recent Omicron Xbb1.5, in MEF$^{TKO-}$ACE2 and MEF$^{T-KO-}$AT cells. Six days after 4OH-TMX or vehicle control treatment, cells were infected with the SARS-CoV-2 Wuhan or Omicron variants B1 and Xbb for 20 h, at an MOI sufficient to infect 20–30% of the control treated cells. The fraction of infected cells was determined by immuno-fluorescence staining and image analysis. As opposed to the strong reduction of infection in cells that did not express TMPRSS2 (Fig 8A, MEF$^{TKO}$ACE2, 4OH-TMX, DMSO), and despite the low levels of TMPRSS2-GFP expression that was detectable by FACS, the expression of TMPRSS2 rescued the infection of the Wuhan strain in dynamin-depleted cells compared to MEF$^{TKO}$ACE2 cells (Fig 8D, MEF$^{TKO}$AT, 4OH-TMX, DMSO). In addition, the combination of dynamin depletion and treatment with apilimod resulted in a complete block of infection of the Wuhan strain in cells that did not express TMPRSS2 (Fig 8A, MEF$^{TKO}$ACE2, 4OH-TMX, Apil. 1 µM). This strong inhibition was significantly rescued by the expression of TMPRSS2 (Fig 8D, MEF$^{TKO}$AT, 4OH-TMX, Apil. 1 µM). Thus, TMPRSS2 allows SARS-CoV-2 Wuhan infection in a dynamin-independent manner. Whether this infection is mediated by direct fusion of viruses at the PM or by dynamin-independent endocytosis remains to be established.

The infection of dynamin-depleted cells by the Omicron variants, on the other hand, was not rescued by TMPRSS2 expression (compare Fig 8B–8C and 8E–8F, 4OH-TMX). Moreover, the levels of infection remained equally low in dynamin-depleted cells treated with Apilimod, regardless of TMPRSS2 expression (compare Fig 8B–8C and 8E-F, 4OH-TMX, Apil. 1 µM). These results confirm that in this model system the omicron lineage viruses do not use TMPRSS2 as efficiently as the ancestral SARS-CoV-2.

## Conclusions

While inhibiting one endocytic pathway results in the block of internalization for receptors such as TfR and its ligands Tf and CPV or ACE2 and furin cleavage resistant S protein trimers of SARS-CoV-2, many viruses can enter cells and initiate infection using more than one receptor and endocytic mechanism (S6 Fig). For viruses partially sensitive to dynamin depletion, such as influenza and alphaviruses, the inhibitory effect was more evident at low MOI. Increasing the virus dosage per cell gradually restored the infection. This could suggest that dynamin depletion blocks internalization of the main viral receptor, and the dynamin-independent pathway is mediated by either alternative, less accessible receptor(s), or by unspecific binding to charged molecules at the cell surface that are internalized via a dynamin-independent pathway. In support of this scenario, previous work has shown that over-expression of temperature sensitive dynamin mutants, which at the restrictive temperature act dominant-negatively, increase the dynamin-independent uptake of fluid phase markers [78]. Similarly, in the dynamin depleted MEF cells used here, fluid phase uptake has been shown to increase [34,72]. Another possibility is that the function of dynamin could involve the clustering of the receptors and thereby tune receptor affinity / avidity for the ligand.

The mechanism and cellular factors involved in the dynamin-independent endocytic pathway(s) mediating infection of the viruses presented in this study remain to be characterized. Identifying the endocytic pathway responsible for SARS-CoV-2 entry will be of particular importance in cells where infection occurs from endosomes, such as human neurons [33,79]. In addition, whether the SARS-CoV-2 co-receptor NRP1, which when co-expressed with ACE2 and TMPRSS2 enhanced virus entry and infection [73,69], contributes to the mechanisms of virus endocytosis is an interesting subject for further studies, in particular, since

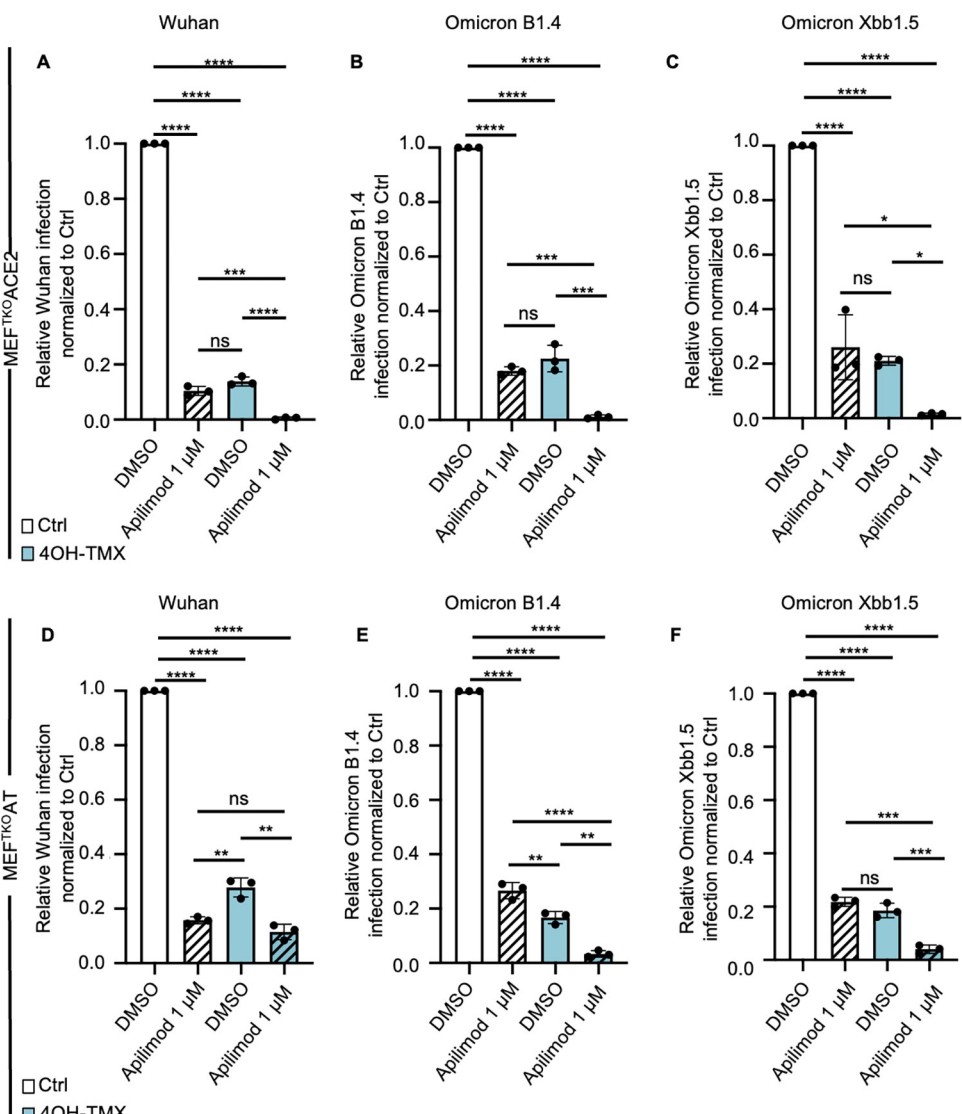

**Fig 8. TMPRSS2 expression bypasses the SARS-CoV-2 Wuhan but not Omicron lineage virus need for dynamin-dependent endocytosis and endosome maturation.** A-C) Image analysis quantification of the levels of SARS-CoV-2 Wuhan (A), Omicron B1.4 (B), and (C) Omicron Xbb1.5 infection following immunofluorescence detection of viral NP protein in MEF$^{TKO}$ACE2 cells. Cells were pre-treated with vehicle (Ctrl) or 4OH-TMX for 6 days and then further treated with DMSO or 1 µM Apilimod 30 min prior to infection. For each variant, the amount of virus was calibrated to infect approximately 20% of the Ctrl + DMSO treated cells at 20 hpi (here normalized to 1). D-F) The above mentioned viruses and drugs were used to infect MEF$^{TKO}$ACE2 cells stably expressing low levels of TMPRSS2-GFP (MEF$^{TKO}$AT) as described for A-C. All values represent the mean and standard deviation of three independent experiments. Cells were infected using an equivalent MOI 3 (virus titers determined in VeroE6-TMPRSS2 cells). More than 10.000 cells were analysed for each experiment after high content imaging and automated image analysis. Statistical analysis was performed using ordinary two-way ANOVA multiple comparisons test (* $p<0.05$; ** $p<0.01$; *** $p<0.001$; **** $p<0.0001$; ns = non significant).

NRP1 is reported to be internalized via both clathrin-dependent endocytosis involving SH3BP4 and GIPC1 binding [80], but also dynamin-independent endocytosis [81,82].

Currently, the dominant SARS-CoV-2 variants are of the Omicron lineage. Recent evidence has shown that this variant favours either endosomal cathepsin-[83] in cell lines or membrane type-matryx metallo proteinases (MT-MMPs) mediated entry rather than TMPRSS2-mediated

infection in human nasal epithelial cells grown at Air-liquid interphase (ALI) [84,85]. This recent work demonstrated that inhibition of endolysosomal acidification/trafficking by bafilomycin A1 does not interfere with Omicron BA.1 infection of primary nasal ALI cells, while it does block Wuhan and Delta variant infections [85]. Meanwhile, three different MMP inhibitors inhibit BA.1 in these primary cells [85]. These findings indicate that it is unlikely that Omicron requires endosomal cathepsins for its entry in the nasal epithelium. Rather, the evidence strongly supports the involvement of MMPs at or near the cell surface [85]. Our results independently indicate that both the B.1 and the Xbb Omicron lineages do not use TMPRSS2 as efficiently as the ancestral Wuhan strain in an *in vitro* model. However, other studies have shown Omicron dependence on serine-proteases both in human nasal and lung epithelial organoid models [86,87], as well as in *in vivo* rodent models [88]. Whether this discrepancy of results is due to specific virus strains or differences in the *in vitro* and *in vivo* infection models remains to be clarified.

Whether fusion of SARS-CoV-2, either mediated by transmembrane proteases such as TMPRSS2 or MT-MMPs [84,85], occurs directly at the cell surface or on early/recling endosomes, and therefore requires endocytosis, remains to be determined. The apical side of polarized epithelial cells, the primary target of SARS-CoV-2 infection, is characterized by a thick layer of cortical actin [89]. Even if cell surface proteases could in principle cleave and activate the viral spikes at the plasma membrane, endocytosis could still be beneficial to cary the activated virion through the cortical actin cytoskeleton and deliver the incoming viral gemone to ribosome enriched cytoplasmic sites. In addition, the outer layer of human nasal mucosa is mildy acidic, an environment that has been recently shown to favour fusion independently of endosomal acidification [90]. While the exact site of viral fusion in ciliated epithelia remains to be accurately determined for SARS-CoV-2, our results show that multiple viruses can infect their target cells using at last two entry mechanisms. While it is, at present, unclear to what extent our findings can be extrapolated to human cells/tissue, inhibiting both entry pathways, PM-fusion and endocytosis-mediated infection, may provide effective antiviral therapies.

## Materials and methods

### Ethics statement

The study was conducted according to the guidelines of the Declaration of Helsinki and was approved by the ethical committee of the Department of Medicine at the Helsinki University Hospital (HUS/853/2020, approved 25 March 2020). Verbal informed consent was obtained from all subjects involved in the study.

### Cells

MEF cells monolayers were grown in high-glucose Dulbecco's Modified Eagle Medium (DMEM; Merk, cat. no. D6546) containing 100 IU/ml of streptomycin and penicillin (Merk, cat. no. P0781), and 10% heat inactivated fetal bovine serum (FBS; Thermo Fisher Scientific, cat. no. 10270–106) in a 37˚C incubator with 5% $CO_2$. Dynamin 1, 2 knockout $MEF^{DKO}$ and dynamin 1, 2, 3 knockout $MEF^{TKO}$ cells were a kind gift of Dr. Shawn Ferguson and Pietro De Camilli [34,72]. The depletion of dynamins in these cells was induced by incubating cells with 3 μM 4OH-TMX (dissolved in EtOH) for two consecutive days. The medium was then removed, and cells grown in fresh medium containing 0.3 μM 4OH-TMX. At day 5 post treatment, the medium was removed, and cells seeded in 96-well imaging plates (PerkinElmer, cat. no. 6005182), at a density of 15,000 (vehicle) or 30,000 (4OH-TMX) cells per well. To obtain equal cell densities in the study, seeding a double number of dynamin-depleted cells was necessary because complete dynamin depletion blocks cell division after day 5 post 4OH-TMX

treatment. For colocalization of soluble trimeric spike with Rab5-EGFP, cells were seeded on fibronectin-coated glass coverslips in 24 well plates at a dilution of 50,000 (vehicle) or 100,000 (4OH-TMX) cells per well. Experiments were performed at day 6 post 4OH-TMX treatment.

To generate MEF$^{DKO}$ transiently expressing human ACE2 and Rab5-EGFP, cells were electroporated with a Biorad Gene Pulser Xcell system, using PBS and 2 μg of plasmid DNA (plenti-6.3-hACE2 [73], pRab5-EGFP [71]). For CPV infections, MEF cells were transfected with feline TfR1 plasmid (pCTfR, 2.5 μg) [40] by using TransIT-LT1transfection reagent (Mirus Bio, WI, USA). To generate MEF$^{TKO}$ cells stably expressing hACE2, cells were transduced with the respective lentivirus (pWPI-IRES-Puro-Ak-ACE2, kindly provided by Dr. Sonja Best, Addgene viral prep # 154985-LV) at MOI 0.3. At 48 hpi, positively transduced cells where selected with puromycin (1 μg/ml) for 5 days and the resistant cells amplified and stored in liquid nitrogen.

To generate MEF$^{TKO}$ACE2 expressing TMPRSS2-EmGFP cells were seeded (1x10$^6$/well) in a 6-well plate and grown overnight in DMEM media with 10% FBS, 2 mM l-glutamine, 1% penicillin-streptomycin. Each well received 200 μl of a solution containing 3x10$^5$ infectious units of a commercially available third-generation lentivirus expressing the gene of human *TMPRSS2* with a C-terminally fused histidine tag followed by the EmGFP protein, and from a separate promoter the puromycin resistant gene (Angio-Proteomie, catalogue number vAP-0101). Lentiviral infections were carried in Dulbecco's modified eagle's medium (DMEM), 2 mM glutamine, 0,5% bovine serum albumin (BSA), and 1x penicillin/streptomycin antibiotic mix, at an MOI of 0.3 infectious units/cells (the lentivirus titre was provided by the manufacturers). After selection for 7 days with 3 mg/ml puromycin, cells were amplified for three passages and then then FACS sorted in high, medium, and low TMPRSS2 expressing cells. Only the low TMPRSS2-EmGFP expressing cells attached to the cell culture dishes, and were used in the study.

## Viruses

All experiments with wild-type or mutant SARS-CoV-2 were performed in BSL3 facilities of the University of Helsinki with appropriate institutional permits. Viruses isolated from consensual COVID-19 patients (permits HUS/32/2018 § 16) were propagated once in Ver-oE6-TMPRSS2 cells, titrated by plaque assay, and stored at -80°C in DMEM, 2% FBS, and 1x Penicyllin/Streptomycin. All SARS-CoV-2 viruses were sequenced by next generation sequencing at the University of Helsinki. SFV-EGFP [91], SINV-mCherry [92], VSV-EGFP [48] and UKNV [93] have been described and were propagated for 22 h in BHK 21 cells in DMEM, non-essential amino acids, 1xGlutamax, 1x Penicyllin/Strepomycin, and 20 mM HEPES pH 7.2. VACV-EGFP [11] was propagated in Hela cells. Influenza A X31 strain was propagated in A549 cells [94]. CPV type 2 was grown, isolated and concentrated as previously described [40]. AdV-C5_EGFP is an E1/E3 deletion mutant virus with E1 region replaced by the enhanced green fluorescent protein (EGFP) gene under the control of cytomegalovirus major immediate early promoter [56]. The virus was grown in 911 cells and purified on CsCl gradients as previously described [95,96]. RV-A1 stock virus was produced as previously described [95].

## Small molecule inhibitors and protein-labelling reagents

All small molecule inhibitors were purchased from Tocris except for Lat-A, Lat-B and 4OH-TMX (Sigma). Drugs were solubilized in DMSO and stored at -20°C, were used at the following concentration unless otherwise indicated in the result session: DMSO 0.1%, Bafilo-mycin A1 25 nM, NH$_4$Cl 20 mM, sucrose 0.45 mM, Chlorpromazine 10 μM, Pitstop 30 μM,

Dyngo-4a (i.e. OH-Dynasore) 18 μM and 30 μM, Dynole 20 μM, Dynasore 80 μM, Cytochalasin-D 30μg/ml, Lat-A 3 μg/ml and Lat-B 3 μg/ml, EIPA 80 μM, Wortmannin 500 nM, ML142 10 μM. Tf-A647 (5 μg/ml, Thermo Fisher Scientific, cat. no. T23366) and CTB-A647 (0.05– 1 μg/ml, Thermo Fisher Scientific, cat. n. C34778). For actin filament labelling, Alexa Fluor 647 Phalloidin was used (ThermoFisher, cat. no. A22287). To fluorescently label the soluble trimeric SARS-CoV-2 spike, provided by Dr. Paul Young and Dr. Dan Watterson [67,68], we used Atto-550 NHS-ester labelling reagent, (Atto-Tec GmbH, cat.n. AD550-35) according to manufacturer instructions for one hour at room temperature in the dark. To remove the excess unbound fluorophore the mixture (500 ml) was passed through NAP-25 Sephadex (GA Healthcare) columns and eluted in 750 μl of PBS according to manufacturer instructions.

## Fluorescence-activated cell sorting (FACS) analysis

After infection, cells in 24-well plates were washed once in PBS and the cells were detached by incubation in 300 μl of 0.25% Trypsin/EDTA solution for 10 min at room temperature. Trypsin was blocked by addition of 300 μl of DMEM containing 10% FBS and cells were then fixed by addition of 600 μl of 8% PFA for 20 min at room temperature in a rotator. The fixation reaction was blocked by addition of $NH_4Cl$ to a final concentration of 20 mM. Cells were centrifuged at 1000 g for 10 min, and the cell pellet resuspended in PBS. In the case of viruses that expressed fluorescent reporter proteins (i.e. SFV-EGFP, VSV-EGFP, VACV-EGFP, and SIN-mCherry), infected cells were directedly analysed by FACS. In the case of CPV, IAV and UUKV virus, cells were permeabilized with 1% saponin (freshly prepared) in PBS containing 0.5% BSA and incubated with primary and secondary antibodies in the presence of 0.5% saponin and 0.5% BSA in PBS. Antibodies to detect CPV NS1 [40], IAV NP [94], and Uukuniemi virus E [93] proteins have been described before. Samples were analysed for viral antigens fluorescence (or viral induced EGFP or mCherry) by a FACS ARIA (BD Biosciences, NJ, USA) and FACSCalibur flow cytometers (BD Biosciences) systems. Non-infected, drug treated cells were used to set the minimal threshold fluorescence levels (the 'gates') during FACS analysis.

## AdV-C5 and RV-A1 infections and immunofluorescence in MEF<sup>DKO</sup> cells

MEF<sup>DKO</sup> cells were seeded into 25 cm$^2$ flask in growth medium containing 2 μM 4OH-tamoxifen (Sigma, cat. no. H-6278) for 48 h. Medium was replaced by growth medium containing 300 nM 4OH-tamoxifen and after two days, cells were trypsinized and seeded into 96-well imaging plates (Greiner Bio-One, cat. no. 655090). Cells were further incubated in medium containing 300 nM 4OH-tamoxifen for 48 h and then incubated with two-fold dilutions of AdV-C5-EGFP stock virus in DMEM medium (Sigma-Aldrich, cat. no. D6429) supplemented with 7.5% fetal calf serum (Gibco/ Thermo Fisher Scientific, cat. no. 10270106), 1% non-essential amino acids (Sigma-Aldrich, cat. no. M7145) and 1% penicillin-streptomycin (Sigma-Aldrich, cat. no. P0781). After 22 h, cells were fixed with 3% PFA in phosphate-buffered saline (PBS) for 30 min at room temperature, quenched with 25 mM $NH_4Cl$ in PBS for 10 min and nuclei were stained with 4',6-diamidino-2-phenylindol (DAPI, Sigma, cat. no. D9542; 1 μg/ml solution in PBS/0.1% TritonX-100) for 20 min. The plate was imaged with a Molecular Devices automated ImageXpress Micro XLS as previously described[2] using 10 × SFluor objective (NA 0.5). Images were analyzed using CellProfiler [97] and mean EGFP intensity over the DAPI mask was determined. Non-infected control wells were used to determine the threshold for an infected cells, threshold being the mean of maximum values from non-infected cells. Knime Analytics Platform (https://www.knime.com/knime-analytics-platform) was used to calculate infection indexes, i.e. fraction of EGFP-positive cells over total number of cells analyzed. RV-A1 infection assay was carried out by incubating cells with two-fold dilutions of RV-A1

stock virus in plain DMEM at 37˚C for 24 h. Subsequently, cells were fixed and processed for immunostaining with mouse anti-VP2 (RT16.7) and secondary goat Alexa Fluor 488-conjugated anti-mouse antibodies (Thermo Fisher Scientific, cat. no. A-11029) as previously described[2], [98]. The plate was imaged with a Molecular Devices automated ImageXpress Micro XLS with 20 × SFluor objective (NA 0.75). Images were analyzed using CellProfiler and mean antibody signals were detected over a cell area corresponding to the DAPI mask extended by three pixels. Non-infected control wells were used to determine the threshold for an infected cell, threshold being the 90% cut-off value from the noninfected cells. Knime Analytics Platform was used to calculate the infection indexes.

## Immunofluorescence assay for CPV, UUKV and IAV

Cells fixed in 4% PFA for 20 min at room temperature, incubated with a blocking solution of PBS containing 20 mM $NH_4Cl$, for 20 min and then permeabilized with 0.1% Triton X-100 in PBS for 10 minutes at room temperature. After two washes with Dulbecco PBS, cells were incubated with primary antibodies against viral proteins for 2 h at room temperature. After two washes in Dulbecco PBS, cells were incubated with fluorescently conjugated secondary antibodies containing 1μg/ml DAPI for 1 h at room temperature. After three washes in PBS, cells in 96-well imaging plates were processed for imaging. Cells in coverslips were rinsed with milli-Q water and mounted on glass slides using Prolong Gold anti fade mounting medium (Thermo Fisher Scientific, P101144) and stored at 4˚C in the dark until imaging.

## Confocal imaging and colocalization analysis

Samples on coverslips (i.e. internalization of SARS-CoV-2 soluble trimeric spike) were imaged with a W1 Yokogawa Spinning Disk confocal microscope using a 60x oil immersion objective. Images were acquired with a Hamamatsu ORCA-Flash4.0 V2 sCMOS (photon conversion factor 0.46) camera. Colocalization was performed with Imaris software to quantify the fraction of internalized spike particles colocalizing with the endocytic markers Rab5-EGFP and Tf-A647. For each three-dimensional confocal stack, spike particles and endocytic vesicles were automatically detected using the Imaris spot colocalization tool as previously described [41]

## Virus entry by sequential immunostaining

The extent of virus internalization in dynamin-depleted and control $MEF^{TKO}$-ACE2 cells was determined by immunofluorescence staining followed by confocal imaging and image analysis. Viruses (equivalent to MOI 50 for SARS-COV-2 and MOI 100 for SFV) were added to cells in 96-well imaging plates (PerkinElmer, CellCarrier 96 ultra, cat. no. 6055302) for SARS-CoV-2 or on glass coverslips for SFV, at 37˚C for indicated times, in the presence of 50 μM cycloheximide to prevent viral protein synthesis. After fixation, cells were processed for sequential immunofluorescence staining to distinguish viral particles on the outer leaflet of the PM, i.e. particles not yet internalized, from internalized virions. Non-internalized viruses were immunostained in fixed cells (4% PFA, 10 min, room temperature) before permeabilization, using a combination of polyclonal antibodies against the viral spike protein (1:200 dilution of anti S1 RBD cat. no. 40592-T62, and 1:200 dilution of anti S2 cat. no. 40590-T62, both from Sino Biological, and E1-E2 antibodies for SFV, kind gift of Prof. Ari Helenius, ETH Zurich), for 3 h at room temperature. After 3 washed with PBS supplemented with 0.5% BSA (PBS-BSA), cells were incubated with secondary antibodies conjugated to an Alexa Fluor 488 fluorophore (1:500 dilution of goat anti rabbit Alexa Fluor 488; Thermo Fisher Scientific, cat. no. A32731), for 2 h at room temperature. After 2 washes with PBS-BSA, and a second fixation with 4% PFA for 5 min at room temperature, cells were washed 3 times with PBS-BSA, and

permeabilized with 0.1% Triton-X 100 for 10 min at room temperature. After 3 washes in PBS-BSA, cells were stained again with the same anti-spike antibodies (dilution 1:200 SARS-CoV-2, dilution 1:1000 SFV E1E2) for 3 h at room temperature. After three washes in PBS-BSA, cells were incubated with DNA staining Hoechst (SARS-CoV-2) (Thermo Fisher Scientific, cat. no. H3569) or phalloidin-568 (Thermo Fisher Scientific) for SFV and secondary antibodies conjugated to a different fluorophore, Alexa Fluor 647 (1:500 dilution of goat anti rabbit Alexa Fluor 647; Thermo Fisher Scientific, cat. no. A32733), for 2 h at room temperature. After 3 washes in PBS, cells were imaged using a Yokogawa spinning disk confocal CQ1 microscope equipped with a 40x air objective (SARS-CoV-2) or with a Leica SP5 equiped with a 63x oil objective. Automated image analysis to detect particles stained by both fluorophores (i.e. particles at the cell surface) and particles stained only by the second fluorophore (i.e. particles inside the cell) was performed with the Yokogawa Cell Path Finder software (SARS-CoV-2) or the Imaris software (for SFV). The E1a-1 antibody, used at dilution 1:1000, was a kind gift of Professor Margaret Kielian (Department of Cell Biology, Albert Einstein College of Medicine, Bronx, New York 10461).

## SFV infection in MEF$^{TKO}$ and drug treatments

For the experiments with SFV-GFP in MEF$^{TKO}$ cells were grown in DMEM supplemented with 10% FBS, 2 mM l-glutamine, 1% non-essential amino acids, and 1% penicillin-streptomycin. Five days after vehicle (EtOH) or 4OH-TMX treatment, cells were seeded at a density of 10,000 cells per well in 96-well imaging plates (PerkinElmer, catalog number 6005182;) and experiment carried at day 6. Inhibitors (30 uM OH-dynasore, Tocris, catalog number 5364) or respective DMSO control, and heparin (Stem Cell Technologies, catalog number 07980) and respective PBS control, were added to cells 30 minutes before infection. SFV-GFP was added on top of wells. After 2h, the medium was replaced with fresh cell-growth medium containing 20 mM NH4Cl pH 7.2. Infections were carried out for 8 h at 37˚C with 5% CO2. Cells were then fixed with 4% paraformaldehyde in PBS for 20 min at room temperature. The GFP signal was enhanced by staining with GFP antibody (Invitrogen, catalog number PA5-22688) and counterstained with Alexa Fluor 488-conjugated goat anti-rabbit secondary antibody (Invitrogen, catalog number A11008). Nuclear staining was done using Hoechst DNA dye. Automated fluorescence imaging was done using a Molecular Devices Image-Xpress Nano high-content epifluorescence microscope equipped with a 10× objective and a 4.7-megapixel CMOS (complementary metal oxide semiconductor) camera (pixel size, 0.332 μm). Image analysis was performed with CellProfiler-4 software ([www.cellprofiler.org](www.cellprofiler.org)). Automated detection of nuclei was performed using the Otsu algorithm built into the software. To automatically identify infected cells, an area surrounding each nucleus (5-pixel expansion of the nuclear area) was used to estimate the fluorescence intensity of the virally expressed EGFP, using an intensity threshold such that <0.01% of positive cells were detected in non-infected wells.

## Western blotting

MEF$^{DKO}$ cells, either control or 4OH-TMX treated, were harvested with same number of cells on day 6 to day 9 of the 4OH-TMX (or EtOH as vehicle control) treatment. MEF$^{DKO}$ cell lysate was mixed with same volume of 2 × Laemmli sample buffer, boiled for 5 min at 95˚C, and run on 10%-15% SDS-PAGE gels. Wet transfer was then conducted on ice with a PVDF transfer membrane (Immobilon-P, IPVH-00010), following western blotting with the mouse anti-Dyn1,2 antibody (BD Bioscience, cat. no. 610245). β-actin was used as loading control to show the overall actin express level was not changed with 4OH-TMX treatment, with mouse anti–β-actin antibody (Abcam, cat. no. ab6267).

## Electron microscopy

SFV and SARS-CoV Wuhan entry was studied using transmission electron microscopy (TEM) analysis in MEF$^{DKO}$ (SFV) and MEF$^{TKO}$ACE2 (SARS-CoV-2) cells. Cells were grown on fibronectin-coated glass coverslips and treated with 4OH-TMX or vehicle (EtOH) control for 6 days and, following virus adsorption at 4°C for 2 h, cells were shifted at 37°C for 15 minutes to allow virus entry. Coverslips were then fixed and processed for flat embedding as described earlier [99]. Thin sections (60 nm) were cut parallel to the coverslip and imaged with Jeol JEM-1400 microscope operated at 80 kV. Images were acquired with Gatan Orius SC 1000B camera from approximately same depth of the cells, capturing 120 images randomly from both conditions each time a virus was observed. The distribution (% of total) of SFV or SARS-CoV-2 Wuhan in CCPs (identified based on clathrin coated endocytic profile), in NPCs (non-coated profiles identified based on absence of clathrin coat) and LEPs (large endocytic profiles) was then counted manually and values are presented as average ±SD.

## Supporting information

**S1 Fig. Tf uptake and cell binding reveal that a small fraction of MEF$^{DKO}$ cells do not respond to 4OH-TMX treatment.** A) Representative fluorescence images of three independent experiments where 20 min uptake of 5 μg/ml Tf-A647 at 37°C was monitored in MEF$^{DKO}$ cells pre-treated with EtOH vehicle control (Ctrl) or 4OH-TMX for 6 days. Before fixation, cells where washed once with growth media at pH 5.5 followed by media at pH 7.4 to remove non internalized Tf from the cell surface (acid wash). Scale bar = 5 μm, nuclei stained with Hoechst (red). B) The white boxed area from panel A was expanded to distinguish cells where the uptake of Tf-A647 was blocked (KO; i.e. the 4OH-TMX-induced KO of dynamins was efficient) from cells that did not respond to 4OH-TMC and internalized levels of Tf-A647 comparable to those of vehicle Ctrl treated cells (indicated as WT in the image). Scale bar = 10 μm, nuclei stained with Hoechst (red). C) Quantification of Tf-A647 uptake shown in A using automated image analysis. A threshold of Tf-A647 fluorescence intensity in the perinuclear area was set to distinguish wild type from KO cells. Values represent the mean of three replicas. Error bars represent the STDEV. Statistical analysis was performed using an unpaired double tailed t-test (ns = non significant). D) Tf uptake was performed as described in A except cells were not acid washed before fixation. Scale bar 100 μm. E) Quantification of Tf-A647 fluorescence per cell after image analysis (a.u. = arbitrary units). Values represent the mean and standard deviation of three independent experiments. Statistical analysis was performed using ordinary two-way ANOVA multiple comparisons test (* $p<0.05$; ** $p<0.01$).
(TIF)

**S2 Fig. Infection of CPV in dynamin1,2-depleted MEF$^{DKO}$ cells is strongly inhibited while SFV can still infect the cells but is inhibited by Heparin.** A) Representative confocal fluorescence images of MEF$^{DKO}$ cells infected with CPV for 24 h. Infected cells are visualized by immunofluorescence using an antibody against the viral non-structural protein NS1. Nuclei are visualized with DAPI DNA stain. Scale bar = 5 μm. The % of infected cells quantified after automated image analysis is shown in the graph. Statistical analysis was performed using a two tailed T-test (**** $p<0.0001$). B) Representative confocal fluorescence images of MEF$^{DKO}$ cells transiently overexpressing the feline TfR and infected with CPV for 24 h after 6 days treatment with EtOH vehicle control (Ctrl) or 4OH-TMX to induce dynamin depletion. Infected cells are visualized by immunofluorescence using an antibody against the viral non-structural protein NS1. Nuclei are visualized with DAPI DNA stain. Scale bar = 5 μm. C) Representative fluorescence image of MEF$^{DKO}$ 6 days after treatment with EtOH vehicle control (Ctrl) or

4OH-TMX and treated with PBS control (no Heparin) or increasing concentrations of Heparin and subsequentely infected for 7 hours with SFV-EGFP (magenta). Nuclei are stained with Hoechst DNA dye (cyan). Scale bar 100 μm. D) Quantification of the experiments shown in C by automated image analysis. Values represent the mean and standard deviation of the mean of three independent experiments. Values are normalized to the % of infected cells obtained in control (Ctr) and vehicle samples (i.e. no heparin, indicated as 1).
(TIF)

**S3 Fig. Dynamin depletion enhances infection of AdV-C5 and RV-A1 in MEF<sup>DKO</sup> cells.** A) Western blot analysis of dynamin 1,2 levels in MEF <sup>DNM1,2, DKO</sup> cells treated with vehicle control or 40H-TMX for 6 days. Indicated amounts of samples were loaded in the gel before blotting. Tubulin was used as a loading control. The Dyn1,2 antibody used recognizes both dynamin 1 and 2. B-C) Quantification of infection in MEF DNM1,2, DKO cells treated with vehicle control (Ctrl) or 4OH-TMX for 6 days and infected with AdV-C5 for 22 h, and D-E) RV-A1 for 24 h. Virus infection was determined by direct fluorescence imaging of AdV-C5 induced EGFP (C, magenta), and after immunofluorescence staining of RV-A1 VP2 protein (E, magenta). Representative epifluorescence images are shown in C and E. The experiment was performed with two technical replicates (#1 and #2). V1-v5 refer to two-fold dilutions of input virus, n.i. = non infected. Scale bar = 10μm.
(TIF)

**S4 Fig. Drugs used and assessment of cytotoxicity.** A) List of the drugs used for SFV experiment shown in Fig 3. The cellular target of the drug, the concentration used, and the cellular function(s) inhibited by each drug are indicated. B) Quantification of cytotoxicity for the drugs that inhibited SFV infection in Fig 3 in MEF DKO cells pre-treated for 6 days with vehicle (Ctrl) or 4-OH-TMX to deplete dynamin 1 and 2. Cell viability was measured by Cell-Titer Glow assay. The staurosposin derivative UCN-01, known to induce apoptosis, was used at a concentration of 20 mM as a positive control for toxicity. The concentration of all the other drygs is indicated in A. C) Schematic representation of the sequential immunostaining protocol. After fixation and before permeabilization viruses on the surface of cells are immunostained using antibodies against the spike protein (Ab 1) followed by secondary antibodies conjugated to a fluorophore (e.g., excitation 488nm, virus out). After a second fixation and permeabilization, the immunostaining is repeated (Ab2) but using secondary antibodies conjugated to a different fluorophore (e.g., excitation 647nm, virus in). D) Digital detection of viruses immunostained with one phluorofore before permeabilization (virus out) and with a second phluorofore after permeabilization (virus in) as described in C. Image analysis was performed using the Imaris software.
(TIF)

**S5 Fig. Sensitivity of SFV and SARS-CoV-2 infection to inhibitors of dynamin, endosome maturation, and serine proteases.** A) Representative fluorescence images of MEF<sup>TKO</sup> cells pretreated with vehicle control (Ctrl) or 4OH-TMX for 6 days and infected for 7 hours with SFV-EGFP (magenta) in the presence of 30 μM OH-dynasore (OH-dyn) or DMSO control. Nuclei are stained with Hoechst DNA die (cyan). Scale bar 100 μm. B) Quantification by automated image analysis of the experiment described in A with two concentrations of OH-dynasore, 30 and 18 μM,. Values represent the mean and standard deviation of four independent experiments. Data are normalized to the infection levels obtained in Ctrl cells treated with DMSO (indicated as 1). Statistical analysis was performed using ordinary two-way ANOVA multiple comparisons test (*p<0,05; *** p<0,001). C) Representative fluorescence images of MEF<sup>TKO</sup>ACE2 cells 6 days after treatment with vehicle control (Ctrl) or 4OH-TMX. After

fixation, non-permeabilized cells were processed for immunofluorescence detection of cell surface ACE2 (magenta) and actin fibers using phalloidin-A488 (green). Nuclei are stained with Hoechst DNA die (cyan). D) Quantification of the mean intensity of ACE2 fluorescent signal per cell suing automated image analysis. Values represent the mean of three replicas, each including more than 10.000 cells. A.U = arbitrary units. Statistical analysis was performed using an two tailed T-test (*p<0,05). E) Representative fluorescence images of MEF$^{TKO}$ACE2 and MEF$^{TKO}$AT pretreated with nafamostat (25 μM or 2.5 μM), apilimod (2 μM or 0.2 μM) and infected with SARS-CoV-2 Wuhan. F) Quantification by image analysis of the experiment shown in E. Values indicate the mean of at least three independent experiments and the error bars represent STDEV. Statistical analysis was performed using ordinary two-way ANOVA multiple comparisons test (*p<0,05; ** p<0,01; ***p<0,001; n.s. = non-significant). (TIF)

**S6 Fig. Schematic representation of virus entry pathways.** The viruses listed with black font were used in this study. The viruses and toxins in gray font were selected from literature. The inset shows inhibitors that bloked dynamin-independent virus infection and its independence from PI3K and Rho GTPases Cdc42 and Rac1, which makes this entry pathway mechanistically different from micropinocytosis. (TIF)

**S1 Data. contain the original data from which all graphs presented in this work where made.** (XLSX)

## Author Contributions

**Conceptualization:** Ari Helenius, Merja Joensuu, Giuseppe Balistreri.

**Data curation:** Ravi Ojha, Anmin Jiang, Elina Mäntylä, Robert Witte, Arnaud Gaudin, Lisa De Zanetti, Maija Vihinen-Ranta, Jason Mercer, Maarit Suomalainen, Merja Joensuu, Giuseppe Balistreri.

**Formal analysis:** Ravi Ojha, Anmin Jiang, Elina Mäntylä, Robert Witte, Arnaud Gaudin, Lisa De Zanetti, Maija Vihinen-Ranta, Jason Mercer, Maarit Suomalainen, Merja Joensuu, Giuseppe Balistreri.

**Funding acquisition:** Maija Vihinen-Ranta, Jason Mercer, Urs F. Greber, Yohei Yamauchi, Pierre-Yves Lozach, Ari Helenius, Olli Vapalahti, Paul Young, Daniel Watterson, Frédéric A. Meunier, Merja Joensuu, Giuseppe Balistreri.

**Investigation:** Ravi Ojha, Anmin Jiang, Elina Mäntylä, Tania Quirin, Robert Witte, Lisa De Zanetti, Maija Vihinen-Ranta, Jason Mercer, Maarit Suomalainen, Urs F. Greber, Merja Joensuu, Giuseppe Balistreri.

**Methodology:** Ravi Ojha, Anmin Jiang, Elina Mäntylä, Tania Quirin, Naphak Modhira, Robert Witte, Arnaud Gaudin, Rachel Sarah Gormal, Maija Vihinen-Ranta, Jason Mercer, Maarit Suomalainen, Urs F. Greber, Yohei Yamauchi, Pierre-Yves Lozach, Olli Vapalahti, Paul Young, Daniel Watterson, Merja Joensuu, Giuseppe Balistreri.

**Project administration:** Giuseppe Balistreri.

**Resources:** Ravi Ojha, Elina Mäntylä, Naphak Modhira, Robert Witte, Rachel Sarah Gormal, Maija Vihinen-Ranta, Jason Mercer, Maarit Suomalainen, Yohei Yamauchi, Pierre-Yves

Lozach, Ari Helenius, Olli Vapalahti, Paul Young, Daniel Watterson, Frédéric A. Meunier, Merja Joensuu, Giuseppe Balistreri.

**Supervision:** Maija Vihinen-Ranta, Maarit Suomalainen, Urs F. Greber, Ari Helenius, Daniel Watterson, Merja Joensuu, Giuseppe Balistreri.

**Validation:** Ravi Ojha, Anmin Jiang, Elina Mäntylä, Maarit Suomalainen, Merja Joensuu, Giuseppe Balistreri.

**Visualization:** Ravi Ojha, Anmin Jiang, Elina Mäntylä, Maarit Suomalainen, Merja Joensuu, Giuseppe Balistreri.

**Writing – original draft:** Merja Joensuu, Giuseppe Balistreri.

**Writing – review & editing:** Ravi Ojha, Anmin Jiang, Elina Mäntylä, Naphak Modhira, Lisa De Zanetti, Rachel Sarah Gormal, Maija Vihinen-Ranta, Jason Mercer, Maarit Suomalainen, Urs F. Greber, Yohei Yamauchi, Pierre-Yves Lozach, Ari Helenius, Olli Vapalahti, Paul Young, Daniel Watterson, Frédéric A. Meunier, Merja Joensuu, Giuseppe Balistreri.

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
