## [Decision Letter · Decision Letter 0]

1 Dec 2023

Dear Dr. Balistreri,

Thank you very much for submitting your manuscript "Multiple animal viruses, including SARS-CoV-2 and its variants, can use alternative cell entry mechanisms for infection" for consideration at PLOS Pathogens. As with all papers reviewed by the journal, your manuscript was reviewed by members of the editorial board and by two independent reviewers. Both reviewers raised some serious concerns that must be resolved. In light of the reviews (below this email), we would like to invite the resubmission of a significantly-revised version that takes into account the reviewers' comments.

We cannot make any decision about publication until we have seen the revised manuscript and your response to the reviewers' comments. Your revised manuscript is also likely to be sent to reviewers for further evaluation.

Sincerely,

Shan-Lu Liu, M.D.. Ph.D.

Guest Editor

PLOS Pathogens

Guangxiang Luo

Section Editor

PLOS Pathogens

Kasturi Haldar

Editor-in-Chief

PLOS Pathogens

orcid.org/0000-0001-5065-158X

Michael Malim

Editor-in-Chief

PLOS Pathogens

orcid.org/0000-0002-7699-2064

Both reviewers raised some serious concerns that must be resolved.

Reviewer's Responses to Questions

**Part I - Summary**

Reviewer #1: This study investigates the role of dynamin-dependent endocytosis in entry of a broad spectrum of enveloped and non-enveloped viruses. Taking advantage of availability of the conditional double- and triple-dynamin KO MEF cells, the authors compared the ability of diverse viruses to infect these and control cells. Most viruses tested were able to infect dynamin KO cells. Intriguingly, the utilization of a yet poorly characterized dynamin-independent pathway was more efficient at higher MOIs. Infection of dynamin KO cells was more reliant on actin dynamics. In particular, the authors found that SARS-CoV-2 entry into ACE2-expressing cells is heavily dependent on dynamin and actin and this dependence was modestly alleviated by expression of TMPRSS2 that cleaves the Spike protein. Spike cleavage at the S2’ position renders it fusion-competent and may direct the virus entry through a dynamin-independent pathway or enable direct fusion with the plasma membrane.

Overall, the results reported in this manuscript are original and of interest the readership of PLoS Pathogens. Most experiments appear to be expertly performed. However, there are several major issues that reduced enthusiasm for this work. A general concern is that the limitations of the conditional KO approach employed are not properly acknowledged. While the conventional dynamin KD or inhibition strategies suffer from incomplete suppression of dynamin expression and off-target effects, the conditional KO of dynamins that is achieved over the course of 6 days is also likely to perturb the cell physiology and have unanticipated consequences for viral infection. No attempts are made to characterize these cells and assess possible changes in the expression of cognate cellular receptors for the viruses tested in this study.

Reviewer #2: Ohja et al. report on the entry pathways used by multiple animal viruses and find that cell entry can be achieved both in the presence and absence of dynamin isoforms. The authors perform a multitude of experiments, but a cohesiveness of the presentation is lacking. While multiple viruses are used in cell culture experiments involving transfection and inhibitory compounds, only one of them is subjected to ultrastructural analysis. This prevents the extrapolation of results to multiple virus families, which appears to be their intention. I believe that it is interesting that virus inoculum dose could impact whether virus uptake/entry occurs in a dynamin-dependent or dynamin-independent manner, and I believe that more experiments centered around that aspect of the article would have resulted in a better presentation.

**Part II – Major Issues: Key Experiments Required for Acceptance**

Reviewer #1: 1. Cell viability under different treatment conditions, including the relatively harsh treatment with actin depolymerizing drugs, is not measured. This is essential for a proper interpretation of inhibition of viral infection.

2. There is an utter lack of information on the MEF dynKO cells stably expressing ACE2 or both ACE2 and TMPRSS2. Without validation of these cells lines, the results related to SARS-CoV-2 infection are not interpretable. It is mentioned that TMPRSS2-GFP expressing cells were FACS-sorted and only cells the lowest TMPRSS2 expression (no results shown) were used for experiments owing to the lack of adherence of cells expressing higher levels of this enzyme. This raises concerns regarding cell viability, especially after treatment with small molecule inhibitors and the ability to express functional levels of TMPRSS2 in MEF dynKO cells.

3. Related to the previous point, in Fig. S5, SARS-CoV-2 infection of TMPRSS2 expressing cells is only marginally inhibited by the TMPRSS2 inhibitor nafamostat and remains quite sensitive to apilimod (inhibitor of endosome maturation). In addition, TMPRSS2 expression only slightly increased SARS-CoV-2 infection relative to ACE2-only expressing cells. This is highly unusual, given the reported robust effects of TMPRSS2 expression in other cell types. The expression and activity of TMPRSS2 in the constructed MEF dyn KO cells must be examined and reported.

Reviewer #2: Major points:

1. This study implicated various methods to investigate the role of dynamin during entry of various viruses, but with just one type of non-human cell examined, MEF. MEF are not physiologically relevant to many viruses studied in this work, especially SARS-CoV-2. At a minimum, the authors should verify the key conclusions with a more relevent cell line / primary tissue / organoid. The dynamin requirement for the entry of a given virus is likely to vary among cell lines and tissue types. This type of work needs to include primary human tissue of relevance to the virus being studied and should involve the study of endogenous cellular factors that facilitate uptake/entry, rather than requiring overexpression. Concluding on how TMPRSS2 interferes with cell adhesion in MEF cells is a good example of how cell line choice is likely to bias the outcome of experiment and is likely a distraction that is unrelated to SARS-CoV-2 biology in human beings.

2. At this stage in the coronavirus pandemic, it is important that much experimentation be centered on Omicron subvariants. However, the findings produced by the authors here on Omicron entry are unlikely to reflect the nature of Omicron entry into physiologically relevant airway tissue. The authors state that other groups have suggested that Omicron entry occurs in a TMPRSS2-independent, endocytic manner. However, the work in those studies was also mostly performed in irrelevant transformed cell lines. In those respects, the findings reported here on Omicron are neither novel nor interesting. Some references that the authors have failed to cite are those indicating that matrix metalloproteinases can serve as Spike-activating proteases in host cells, especially Omicron lineages. In particular, there is evidence that the entry mechanism of Omicron into primary nasal epithelia is not endocytic but is MMP-dependent (Shi et al. bioRxiv 2023).

3. The title of the article is misleading. The use of the term “alternative” without referencing a canonical entry pathway is inappropriate. I think the authors are trying to state that these viruses use a variety of entry pathways (that does not mean “alternative”). While the use of different viruses in this study seems like an advantage, it complicates the message of the article and dilutes its overall significance. It has already been known for a long time that the viruses tested here use a variety of entry pathways into human cells. The authors need to hone their story to emphasize a point of importance that is both novel and significant.

**Part III – Minor Issues: Editorial and Data Presentation Modifications**

Reviewer #1: 1. The increased use of a dynamin-independent entry pathways at higher MOI can be presented more clearly throughout the paper. Instead of asking the reader to visually estimate the ratio of infection in control and dynamin KO cells, the infection ratio can be plotted to reveal the relative contributions of the two pathways.

2. The existence of an additional, dynamin-independent macropinocytosis-like pathway has been described for the influenza virus (https://doi.org/10.1371/journal.ppat.1001329) – a study that should probably be referenced here.

3. The results shown in Fig. 6 are not readily interpretable. One must “eyeball” the effects of dyn KO and actin depolymerization. To show the LatA effect on control and dyn KO cells, the respective ratios of infection values can be plotted. In some cases, a log scale can be useful. Importantly, it is unclear why infections do not scale linearly with MOI. Minor: The authors interpret the results of Fig. 6 as supporting the ability of SARS-CoV-2 to use dynamin-independent pathway. Given that cell surface proteases other than TMPRSS2 have been implicated in SARS-CoV-2 Spike cleavage, can the authors rule out the possibility of direct fusion with the PM of dyn KO cells.

4. Why rely on the use of apilimod alone to assess the contribution of endocytic entry, when specific inhibitors of cathepsin L and other capthepsins have been widely used to inhibit endocytic entry of SARS-CoV-2?

5. It is unclear if "single-receptor" viruses, like CPV, are indeed exclusively binding to one receptor (e.g., TfR). Most viruses appear to bind non-specifically to the cell surface molecules. This brings us to the enhanced use of dynamin-independent endocytosis by virus at higher MOIs. Unfortunately, the mechanism of such virus input-dependence of entry route is not addressed in this study.

6. Fig. 2: The infections do not scale linearly with virus inputs throughout. I’d consider repeating the experiments using lower MOI.

7. Fig. 3: Negative controls using viruses that, unlike SFV, are resistant to some of the experimental manipulations are lacking. Also, inhibition of SFV infection by EIPA is shown in Fig. 3 but not mentioned or interpreted in Results.

8. No error bars or statistical analysis shown in Fig. S3.

9. The super-resolution PALM results presented in Fig. S4 appear out of place, as the relationship between single actin molecule diffusion coefficients or foci observed by phalloidin staining and the function of actin in the context of endocytosis and virus entry are unclear. Unless the modest changes in single actin molecule mobility in dynamin KO cells and endocytic activity is established, I’d consider removing these results, as these do not add much to understanding of alternative viral entry pathways.

10. Fig. 4: The preference of virus for NCPs in dynamin KO cells appears marginal. However, there is rather strong preference for LEPs in control cells, which is not commented on. How is this observation compatible with virus entry via clathrin- and dynamin-dependent endocytosis?

11. Fig. 5: The experiments measuring uptake of the SARS-CoV-2 spike ectodomain lack controls assessing S binding to ACE2-negative cells and showing the specificity of this binding to ACE2-expressing cells (e.g., competition with RBDs).

Reviewer #2: There are a number of typos and english language/grammar could be improved.

PLOS authors have the option to publish the peer review history of their article (what does this mean?). If published, this will include your full peer review and any attached files.

Reviewer #1: No

Reviewer #2: No
---

## [Decision Letter · Decision Letter 1]

5 May 2024

Dear Dr. Balistreri,

Thank you very much for submitting your manuscript "Dynamin independent endocytosis is an alternative cell entry mechanism for multiple animal viruses" for consideration at PLOS Pathogens. As with all papers reviewed by the journal, your manuscript was reviewed by members of the editorial board and by two independent reviewers. In light of the reviews (below this email), we would like to invite the resubmission of a significantly-revised version that takes into account the reviewers' comments. Some previous reviewer's comments are not directly addressed, along with some new issues.

We cannot make any decision about publication until we have seen the revised manuscript and your response to the reviewers' comments. Your revised manuscript is also likely to be sent to reviewers for further evaluation.

Sincerely,

Shan-Lu Liu, M.D.. Ph.D.

Guest Editor

PLOS Pathogens

Guangxiang Luo

Section Editor

PLOS Pathogens

Michael Malim

Editor-in-Chief

PLOS Pathogens

orcid.org/0000-0002-7699-2064

Reviewer's Responses to Questions

**Part I - Summary**

Reviewer #2: The authors have revised their submission, but an accurate point-by-point response to the reviewers' comments is missing. It is unclear if this is accidental or deliberate, or if the authors were paraphrasing the reviewers' comments in their point-by-point response.

Reviewer #3: This study used the Dyn-KO-MEFs to examine the cell entry dependency on cellular dynamins by a group of large and small animal viruses. The authors concluded that single-receptor viruses are exclusively depend on dynamin for cell entry, whereas multi-receptor viruses can enter cells via both dynamin-dependent and dynamin-independent endocytic pathways. The experiments were properly controlled. Interpretation of the data can be improved, as explained below.

**Part II – Major Issues: Key Experiments Required for Acceptance**

Reviewer #2: The authors should upload a new point-by-point response to reviewers that uses the exact words of the reviewers, as laid out in the PLOS Pathogens decision letter on December 1st, 2023.

Reviewer #3: Virus entry has been studied quite extensively for the past decades. An account of the different endocytic pathways that are exploited by different viruses will help to understand the relevance of the subject investigated in this study. For example, which endocytic pathways are dynamin-dependent or dynamin-independent, and are used by which viruses. In this context, a model at the end of the manuscript to summarize the findings will also be helpful, also highlight why studying dynamin in the context of virus entry is important.

No evidence is provided on which viruses use multiple receptors. For example, will depleting the known receptor of a virus block the entry/infection? Without this type of data, the link between dynamin-independent cell entry and multiple receptor usage cannot be established. One argument can be that at high doses (such as MOI=10 or 100), virus particles can aggregate during the concentration process and can be engulfed by cells in a receptor-independent and dynamin-independent way.

Most of the experiments measure viral protein expression >20 hours after infection, not virus entry per se, this caveat should be acknowledged.

It is stated that (line 228) MEF(DKO) cells stop dividing, how relevant then are the findings made with these KO cells, particularly viral protein expression levels were measured as a readout of virus entry? In another word, in healthy primary cells, when dynamin function is transiently inhibited, how much of the observations in this study can be reproduced? Related, in lines 242-245, the authors stated that CEM and dynamin inhibitors inhibited viruses in both the wild type and Dyn-KO cells; in addition to the off-target possibility, can the authors titrate the drug concentrations to circumvent this possible effect? These inhibitors are widely used in many studies.

**Part III – Minor Issues: Editorial and Data Presentation Modifications**

Reviewer #2: (No Response)

Reviewer #3: Some typos to correct:

line 40, "we studies"

line 86, ref is missing

line 300, "this ration"

line 303, "where in CCP"

line 461, "37 C"

line 522, "The function"

PLOS authors have the option to publish the peer review history of their article (what does this mean?). If published, this will include your full peer review and any attached files.

Reviewer #2: No

Reviewer #3: No
---

## [Decision Letter · Decision Letter 2]

27 Aug 2024

Dear Mr. Balistreri,

Thank you very much for submitting your manuscript "Dynamin independent endocytosis is an alternative cell entry mechanism for multiple animal viruses" for consideration at PLOS Pathogens. As with all papers reviewed by the journal, your manuscript was reviewed by members of the editorial board and by several independent reviewers. In light of the reviews (below this email), we would like to invite the resubmission of a significantly-revised version that takes into account the reviewers' comments.

Please ensure to fully address the reviewer’s comments and we will proceed with a last round of review with the intention of making final decision.

We cannot make any decision about publication until we have seen the revised manuscript and your response to the reviewers' comments. Your revised manuscript is also likely to be sent to reviewers for further evaluation.

Sincerely,

Shan-Lu Liu

Guest Editor

PLOS Pathogens

Sonja Best

Section Editor

PLOS Pathogens

Michael Malim

Editor-in-Chief

PLOS Pathogens

orcid.org/0000-0002-7699-2064

Reviewer's Responses to Questions

**Part I - Summary**

Reviewer #2: In this additional revision, Ohja et al. have responded to most of my concerns, and in some cases, have done so adequately. However, there remain several points that have not been addressed.

Reviewer #3: The authors made great efforts to address comments from the reviewers, and provided as much new data as possible as suggested. The manuscript has been significantly improved.

**Part II – Major Issues: Key Experiments Required for Acceptance**

Reviewer #2: Major:

1. In their rebuttal letter, the authors state that “the role of endocytosis in the SARS-CoV-2 infection of primary respiratory cells has never been directly addressed to date. To our knowledge, available studies are limited to investigating the role of endosomal pH and endolysosomal proteases but not whether the virus requires endocytosis. Similarly, the fusion of viruses at the plasma membrane has been inferred by the ability of viruses to escape inhibitors of endosome acidification and endolysosomal proteases.” This is precisely why I brought up the recent publication Shi et al. Nat Comm 2024 in my peer review report. This study uses inhibitors to indicate the Omicron, in contrast to Delta and SARS-CoV-2 from 2020, enters primary nasal epithelia at the air-liquid interface in a TMPRSS2-independent manner, but also in a cathepsin-independent manner. This suggests that Omicron may enter primary human cells by undergoing fusion at a non-endolysosomal compartment. Supporting this, use of the endolysosomal acidification inhibitor (and inhibitor of endosomal maturation) Bafilomycin A was shown to inhibit Delta and SARS-CoV-2 from 2020, but not Omicron. This suggests that Omicron does not require functional endolysosomal machinery in order to enter primary human epithelial cells. Furthermore, combining MMP inhibitors with Bafilomycin A inhibited Omicron to the same extent as MMP inhibitors alone, suggesting that Omicron likely uses MMP proteases to trigger fusion at the cell surface or early/recycling endosomes close to the cell surface. The authors should discuss these findings, and they can state their limitations (inability to prove cell surface fusion versus early/recycling endosome fusion) while presenting their own data in its context. The authors provide evidence supporting that Omicron infection occurs in a TMPRSS2-independent manner, but this has already been demonstrated by multiple groups (including the Shi et al. Nat Comm 2024 paper mentioned above). Therefore, this is not a new or useful finding in and of itself. The authors should review the literature claiming that Omicron does, or does not, undergo fusion in endosomes in human cells.

2. I stand by the statement that, if we really want to understand the many ways by which human viruses enter human cells, experiments must be performed with human cells (and ideally, primary tissue). Even if it represents a clean genetic background to address dynamin isoforms involved in virus entry, results obtained in MEF cells can’t be used to report “novel findings” about how human viruses enter human cells. These results can’t be used to infer insight into the biology and pathogenesis of human viruses that cause disease in humans. The authors need to discuss the limitations of their own findings at length and state that it is unclear to what extent their findings can be extrapolated to human cells/tissue.

3. I’m not convinced that the authors have properly excluded macropinocytosis as one of the dynamin-independent endocytosis mechanisms at play in their experiments. The fact that EIPA inhibits infection to a certain degree supports that fluid-phase, macropinocytotic entry is taking place. Also, wortmannin is not a suitable inhibitor of macropinocytosis, as it is a pan class I/class III PI3K and mTOR inhibitor. Caution: the authors use the term “micropinocytosis” in this manuscript instead of the proper term macropinocytosis.

Reviewer #3: Dynamin-independent endocytosis and thus dynamin-independent entry of IAV, LCMV, HPV and other viruses have been documented, as highlighted in the 2010 review article (Figure 1, ref 1), this needs to be well presented in the Introduction so that the novelty and importance of this study can be justified, i.e. contributions to what is already known in the literature.

MEFs are not the natural target cells of many viruses that were investigated in the study, some of the findings thus may not be reproducible for a virus in its primary target cells, this weakness needs to be acknowledged.

Half of results centered on SARS-CoV-2, which can be shortened, some of the results are confirmative in nature, because similar findings have been reported in the literature, such as TMPRSS2-independent entry of the Omicron variant.

**Part III – Minor Issues: Editorial and Data Presentation Modifications**

Reviewer #2: N/A

Reviewer #3: (No Response)

PLOS authors have the option to publish the peer review history of their article (what does this mean?). If published, this will include your full peer review and any attached files.

Reviewer #2: No

Reviewer #3: No
---

## [Decision Letter · Decision Letter 3]

19 Oct 2024

Dear Mr. Balistreri,

Thank you very much for submitting your manuscript "Dynamin independent endocytosis is an alternative cell entry mechanism for multiple animal viruses" for consideration at PLOS Pathogens. As with all papers reviewed by the journal, your manuscript was reviewed by members of the editorial board and by several independent reviewers. The reviewers appreciated the attention to an important topic. Based on the reviews, we are likely to accept this manuscript for publication, providing that you modify the manuscript according to the review recommendations.

Sincerely,

Shan-Lu Liu

Guest Editor

PLOS Pathogens

Sonja Best

Section Editor

PLOS Pathogens

Michael Malim

Editor-in-Chief

PLOS Pathogens

orcid.org/0000-0002-7699-2064

Reviewer Comments (if any, and for reference):

Reviewer's Responses to Questions

**Part I - Summary**

Reviewer #2: (No Response)

Reviewer #3: My critiques have been addressed.

**Part II – Major Issues: Key Experiments Required for Acceptance**

Reviewer #2: The authors have nearly adequately addressed my concerns. However, their is slight revision required to how the authors cite the Omicron literature with regards to endosome/plasma membrane fusion. At line 584 of the Conclusions section, the authors state "Recent evidence has shown that this variant favours either endosomal cathepsin in cell lines or membrane type-matryx (spelling error - matrix) metallo proteinases (MT-MMPs) mediated entry rather than TMPRSS2-mediated infection in human nasal epithelial cells grown at air-liquid interphase (ALI)."

In addition to reference 84 cited here (Chan et al. Science Advances), the authors should include citation for Shi et al. Nature Communications 2024 (PMID: 38291024) as discussed in previous round of review. Also, at line 593, Shi et al. should be cited alongside reference 84. The authors failed to describe how Shi et al. demonstrates that inhibition of endolysosomal trafficking by bafilomycin A1 does not interfere with Omicron BA.1 infection of primary nasal ALI, while it does block WA1 and Delta infection. Meanwhile, three different MMP inhibitors inhibit BA.1 in this primary tissue. These findings indicate that is unlikely that Omicron requires endosomal cathepsins for its entry. Rather, the evidence strongly supports the involvement of MMPs at or near the cell surface.

Furthermore, there are two typos at lines 574-575: please correct to "particular" and "In addition."

Reviewer #3: No more major issues.

**Part III – Minor Issues: Editorial and Data Presentation Modifications**

Reviewer #2: (No Response)

Reviewer #3: None.

PLOS authors have the option to publish the peer review history of their article (what does this mean?). If published, this will include your full peer review and any attached files.

Reviewer #2: No

Reviewer #3: No

Figure Files:

Data Requirements:

Reproducibility:

References:

---

## [Editor Report · Decision Letter 4]

23 Oct 2024

Dear Dr. Balistreri,

We are pleased to inform you that your manuscript 'Dynamin independent endocytosis is an alternative cell entry mechanism for multiple animal viruses' has been provisionally accepted for publication in PLOS Pathogens.

Best regards,

Shan-Lu Liu

Guest Editor

PLOS Pathogens

Sonja Best

Section Editor

PLOS Pathogens

Michael Malim

Editor-in-Chief

PLOS Pathogens

orcid.org/0000-0002-7699-2064
---

## [Editor Report · Acceptance letter]

4 Nov 2024

Dear Mr. Balistreri,

We are delighted to inform you that your manuscript, "Dynamin independent endocytosis is an alternative cell entry mechanism for multiple animal viruses," has been formally accepted for publication in PLOS Pathogens.

Best regards,

Michael Malim

Editor-in-Chief

PLOS Pathogens

orcid.org/0000-0002-7699-2064